# Deep-learning-enabled online mass spectrometry of the reaction product of a single catalyst nanoparticle

Henrik Klein Moberg, Giuseppe Abbondanza, Ievgen Nedrygailov, David Albinsson [ID], Joachim Fritzsche [ID] & Christoph Langhammer [ID] [✉]

Extracting weak signals from noise is a generic challenge in experimental science. In catalysis, it manifests itself as the need to quantify chemical reactions on nanoscopic surface areas, such as single nanoparticles or even single atoms. Here, we address this challenge by combining the ability of nanofluidic reactors to focus reaction product from tiny catalyst surfaces towards online mass spectrometric analysis with the high capacity of a constrained denoising autoencoder to discern weak signals from noise. Using CO oxidation and $C_2H_4$ hydrogenation on Pd as model reactions, we demonstrate that the catalyst surface area required for online mass spectrometry can be reduced by $\approx 3$ orders of magnitude compared to state of the art, down to a single nanoparticle with $0.0072 \pm 0.00086\,\mu m^2$ surface area. These results advocate deep learning to improve resolution in mass spectrometry in general and for online reaction analysis in single-particle catalysis in particular.

The ability to detect and analyze weak signals amidst high levels of noise has led to important discoveries across a myriad of scientific disciplines using widely different experimental methodologies[1–10]. Mass spectrometry constitutes such an experimental method since it enables the detection of very small amounts of particles, and has become a workhorse from (astro)physics to biology to chemistry[11,12]. Accordingly, the method comes in different variants tailored for the species subject to characterization and the operational conditions. Mechanistically, it counts ionized particles, molecules or atoms in vacuum by separating them according to their mass/charge ratio, and thereby determines their (molecular) weight.

In catalysis and surface science, mass spectrometry is widely used to quantitatively analyze the composition of molecules desorbing from surfaces, often from a surface-chemical reaction[13,14]. In this application, usually a quadrupole is used for mass selection due to its high resolution and sensitivity[15,16]. Such quadrupole mass spectrometers (QMS) have been instrumental in the development of our understanding of nanocatalytic processes. Nevertheless, the technical advance of QMS systems used for so-called residual gas analysis has been limited during the last decades, and, therefore, no significant progress has been made in terms of improving their limit of detection and their ability to discern very weak signals from noise[17,18]. Concurrently, this state-of-the-art (SotA) is becoming an increasingly limiting factor since accurately measuring catalytic activity and selectivity from small amounts of precious catalyst materials, such as size- or shape-selected model catalysts[19] or single atom catalysts[20–22] that can only be obtained in small quantities, is crucial for the development of new catalyst materials and for advancing our understanding of catalytic processes. To this end, in the most extreme implementation, the aim of so-called single-particle catalysis is to study catalytic reactions on individual nanoparticles to overcome ensemble averaging effects[23].

The first strategy to enable such experiments employs detection of photon or electron signals that report on either the product molecules formed, on reactant molecules consumed, on the catalyst particle itself, or on temperature changes generated by the reaction[23–28]. In this context, single-molecule fluorescence microscopy stands out due to its ability to resolve individual reaction events on single catalyst nanoparticles with very high spatial resolution[29–31]. While both elegant and effective for the specific reactions chosen, however, these approaches are compatible only with catalysis in the liquid phase and lack the wide and generic applicability of mass spectrometry.

Department of Physics, Chalmers University of Technology, Göteborg, Sweden. [✉] e-mail: clangham@chalmers.se

Furthermore, they often cannot deliver information about both the amounts and molecular weight of the species involved in the reaction.

To address this limitation, miniaturization of the reactor to the level of microreactors[32–34] and even nanoreactors[35] to reduce the catalyst surface area necessary to obtain measurable (QMS) signals has been implemented. Very high sensitivity can be obtained in such systems by dramatically reducing reactor volume and thereby directing the entire gas flow from the catalyst bed to a QMS to maximize the fraction of reaction product available for analysis. In this way, as we recently have demonstrated using a single nanofluidic channel as the catalyst bed, the minimal required active catalyst surface area for online QMS measurements was reduced by ca. 1.5 orders compared to the lower limit proposed for microreactors[32], i.e., down to ca. $10 \ \mu m^2$ active surface area[36].

While this indeed was a significant advance, it is still 2-3 orders away from online QMS measurements from single nanoparticles. To overcome this gap, we employ here a modified type of nanofluidic reactor and combine it with a constrained convolutional denoising auto-encoder. This harnesses the reaction product focusing capability of nanofluidic reactors with the versatility, high-resolution and sensitivity of the QMS, with the power of deep learning to detect and analyze very weak signals hidden in noise. As we demonstrate on the example of CO oxidation, i.e., $CO + \frac{1}{2} O_2 \rightarrow CO_2$, over a Pd model catalyst, this approach leapfrogs the limit of detection of a SotA QMS system by $\approx 3$ orders, and enables online mass spectrometric analysis of reaction product from a single Pd nanoparticle with active surface area $A \approx (0.0072 \pm 0.00086) \ \mu m^2 = (7200 \pm 860 \ nm^2)$. Furthermore, we show on the example of $C_2H_4$ hydrogenation to $C_2H_6$ that this concept can be applied to a wide range of catalytic reactions and to boost the performance of a QMS inferior in terms of intrinsic sensitivity. Finally, we note that the application of machine learning to different modalities of mass spectrometry has a short but lively history[37–44], but the application of machine learning to discern small QMS signals from noise, as we do here, is hitherto completely unexplored.

## Results

The reaction of CO with $O_2$ over Pd catalysts forming $CO_2$ is one of the most studied reactions in catalysis, both due to its practical relevance and due to its role in studies of structure-function correlations and (surface) oxide formation[45,46]. Therefore, we chose it as our model reaction with the aim to enable QMS measurements of the $CO_2$ formed on a single Pd nanoparticle. We have developed a nanofluidic reactor connected to U-shape microfluidic in- and a straight outlet channel(s), which are connected to a macroscopic reactant inlet system operated by mass flow controllers at 3 bar, and to a QMS mounted on a UHV chamber with $1 \cdot 10^{-10}$ mbar base pressure (Fig. 1a and Supplementary Fig. 1). This in- and outlet system is connected to a nanofluidic catalyst bed with $30 \ \mu m \times 10 \ \mu m \times 200 \ nm (L \times W \times H)$ dimensions via a nanofluidic inlet channel ($505 \ \mu m \times 10 \ \mu m \times 200 \ nm$) and an outlet channel with dimensions $100 \ \mu m \times 500 \ nm \times 200 \ nm$ (Fig. 1b, c). This design ensures that the catalyst is operated in the well-mixed regime, where concentration gradients due to reactant conversion are effectively eliminated[35]. Furthermore, we note that the gas flow in such a system is in the slip flow and transitional flow regimes, with the Knudsen number increasing in the mean flow direction throughout the nanofluidic reactor[35]. Consequently, momentum, heat and mass transfer boundary layers surrounding reacting particles are thinner than in the continuum regime, and the mass diffusivities approach the Knudsen diffusivity. In fact, nanofluidic reactors therefore constitute effective "model pores" since the aforementioned effects all are present in technically widely used porous catalyst systems[36]. Using a resistive heater, the nanoreactor zone can be heated up to 450 °C[36]. For this study, we have fabricated four nanoreactor chips with identical gas in- and outlet systems, as well as catalyst beds. Using electron-beam lithography (EBL) nanofabrication[47], they were decorated with (regular arrays of) $n = 1000, 10, 1$ or $0$ Pd particles with diameter $d = 59.4 \pm 3.7$ nm and height $h = 23.5 \pm 1.6$ nm, as SEM image analysis reveals, physical vapor deposited (PVD) onto a 8 nm thick $SiO_2$ support layer that separates them from a chemically inert plasmonic Au nanoparticles previously PVD-grown through the same EBL mask with $d = 97.6 \pm 6.6$ nm and $h = 35.3 \pm 2.3$ nm (Fig. 1c–g and Supplementary Fig. 2 and Methods section on nanofabrication of the nanoreactor chip[36,48]. This corresponds to an active Pd surface area $A \approx (0.0072 \pm 0.00086) \ \mu m^2 = (7200 \pm 860) \ nm^2$ per particle. We chose this particular hybrid nanoparticle design, where the Au element serves as a plasmonic light scatterer (Pd is optically dark[49]), to enable the verification of the presence of the anticipated number of particles in the enclosed nanoreactors using dark-field scattering microscopy (Fig. 1d–g).

### CO oxidation on Pd model catalysts in nanofluidic reactors

To prepare these systems for experiments in reaction environment, we first exposed them to a conditioning sequence, followed by a full CO oxidation sequence at 280 °C (see Methods section on sample mounting and pre-treatment and Supplementary Figs. 3–6). Subsequently, we initiated the experiment sequence consisting of 15 min CO/$O_2$ mixture pulses at a constant 6% reactant concentration and starting with 6% percent CO, separated by 15 min in Ar. Such sequences were executed from 280 to 450 °C in 20 °C steps for the $n = 1000$ sample and 10 °C steps for the $n = 10$ and 1 samples, to sweep the entire $0 \le \alpha_{CO} = P_{CO}/(P_{O_2} + P_{CO}) \le 1$ range in steps of 0.05, where $\alpha_{CO}$ is the relative CO concentration in the gas mixture. (Fig. 2a–c). At each new temperature, a single 30 min pulse of 4% $O_2$ and 2% CO in Ar carrier gas, followed by a 15 min pure Ar pulse, was applied to reset the state of the catalyst.

Focusing first on the highest reaction temperature of 450 °C and $n = 1000$ (total active surface area of $A \approx (7200 \pm 860) \ nm^2 \cdot 1000 = (7.2 \pm 0.86) \ \mu m^2$), we observe a QMS response distinctly above the noise floor for all $\alpha_{CO} \in (0, 1)$ values (Fig. 2d). Furthermore, starting from $\alpha_{CO} = 1$, we observe the characteristic reaction rate increase as more $O_2$ is added until the maximum rate is reached at $\alpha_{CO}^{max} = 0.65$. This is very close to the stoichiometric $\alpha_{CO} = 0.66$ (the small difference is the consequence of the 0.05 $\alpha_{CO}$ steps), which indicates that CO poisoning is very mild at this temperature and that mass transport gradients are negligible, as expected for a well-mixed catalyst bed. Upon decreasing $\alpha_{CO}$ beyond $\alpha_{CO}^{max}$, the reaction rate decreases in an essentially linear fashion.

Decreasing the temperature to 340 °C induces both a global decrease in reaction rate and a shift of $\alpha_{CO}^{max}$ to a lower value of 0.5, due to stronger CO poisoning (Fig. 2e). This trend continues upon temperature reduction to 280 °C (Fig. 2f), with a QMS signal still distinctly above the noise floor and $\alpha_{CO}^{max}$ reduced to 0.3. These measurements thus corroborate our experimental setup and approach since they deliver results in agreement with the well-established understanding of the CO oxidation over Pd catalysts[45,46]. Hence, they constitute a relevant baseline as we further reduce the catalyst surface area towards a single Pd nanoparticle.

As the first step, we repeated the experiments for an identical nanoreactor but with $n = 10$, which corresponds to $A \approx (7200 \pm 860) \ nm^2 \cdot 10 = (0.072 \pm 0.0086) \ \mu m^2$ (Fig. 2g–i). We find that at 450 °C, $CO_2$ pulses still are resolved and that the counts is $\approx 2$ orders smaller compared to $n = 1000$. Importantly, however, we also see that the $CO_2$ signal induced by the reactant pulses approaches the noise floor. Accordingly, reducing the temperature makes discerning the $CO_2$ produced increasingly difficult. This becomes clear when extracting the $CO_2$ counts for each $\alpha_{CO}$ pulse and directly comparing the $CO_2$ counts vs. the $\alpha_{CO}$ trend for the $n = 1000$ and 10 samples. At $T = 450$ °C, the trends obtained are qualitatively very similar and thus corroborate that the catalyst is operated at identical conditions (Fig. 2j).

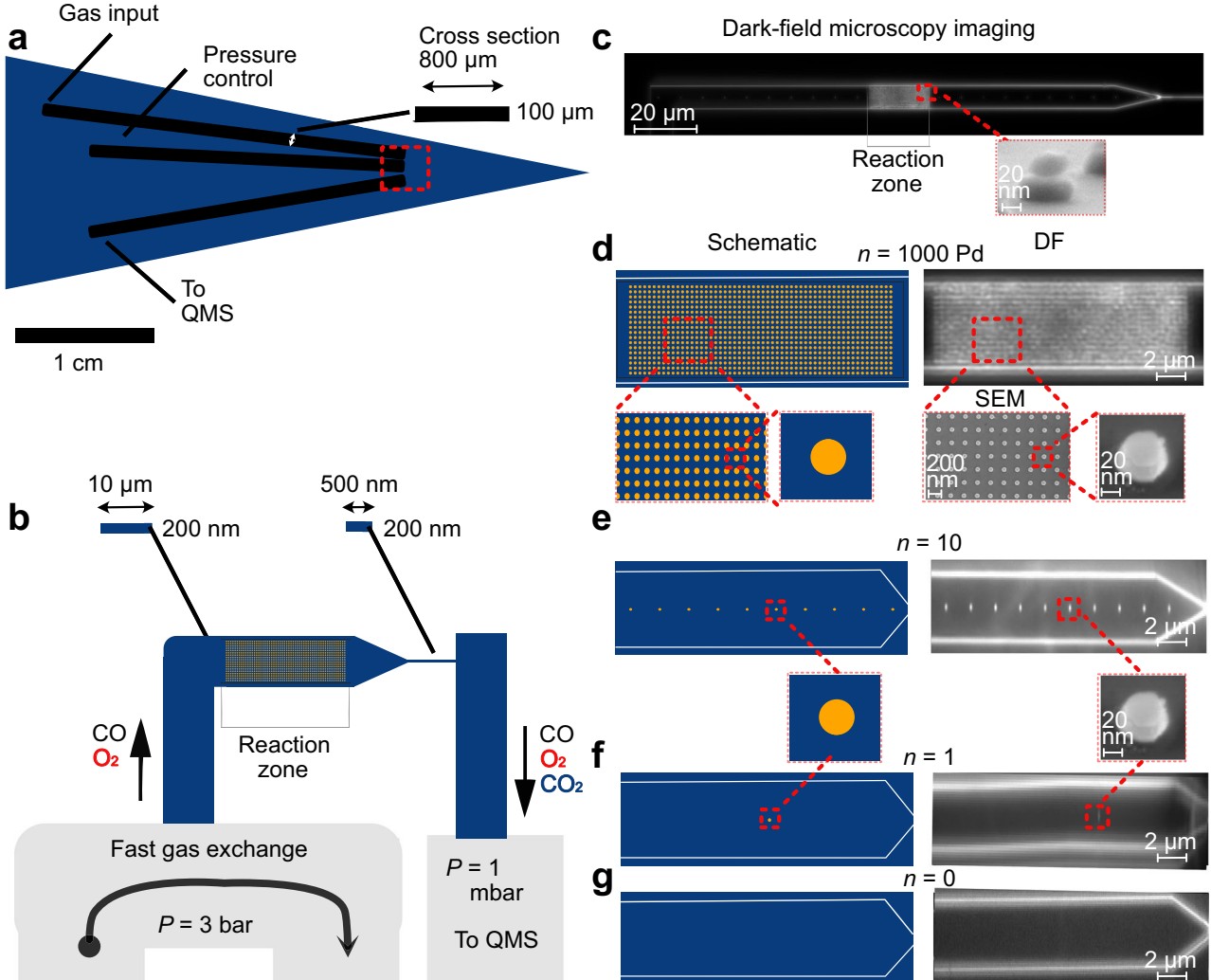

**Fig. 1 | Experimental setup. a** Schematic of the micro- and nanofabricated reactor chip. It is comprised of a U-shaped microfluidic in- and a simple straight outlet system that connects to the model catalyst bed (the nanofluidic reactor with the reaction zone) on either side, as well as to the high-pressure gas handling system and the QMS, respectively. The cross-sectional dimensions of the microfluidic channels are also shown. **b** Schematic depiction of the 30 μm long well-mixed reaction zone (the model catalyst bed that contains the nanoparticle(s)) with cross sectional dimensions 10 μm × 200 nm, as well as its connections to the microfluidic in- and outlet system via a smaller microfluidic inlet channel (also with cross sectional dimensions 10 μm × 200 nm) and via a nanofluidic outlet "capillary" with cross sectional dimensions 500 nm × 200 nm. Note that the schematics are not drawn to scale. **c** Dark-field scattering microscopy image of a nanofluidic catalyst bed containing 1000 nanoparticles arranged in a regular array in the reaction zone. The inset depicts a side-view scanning electron microscopy (SEM) image of the used Au/SiO₂/Pd hybrid nanostructures. Scale bar 100 nm. The catalytically inert Au

element enables single-particle dark-field scattering (DF) imaging to confirm the presence of the correct number of catalyst particles inside the sealed nanofluidic reactor, without itself participating in the reaction[36]. **d** Schematic depiction and zoomed-in DF image of the reaction zone of the nanofluidic catalyst bed used in our experiments, with a regular array of $n = 1000$ nanoparticles that become distinctly visible in the DF image. Also shown are two top-view SEM images of a similar array of nanoparticles, as well as a side-view of a single nanoparticle, prepared on an open surface to facilitate SEM imaging. Scale bar 100 nm. The inset image below "SEM" includes a scale bar for reference. **e** Schematic depiction and zoomed-in DF image of the reaction zone of the catalyst bed containing $n = 10$ Pd nanoparticles used in our experiments. **f** Schematic depiction and zoomed-in DF image of the reaction zone of the catalyst bed containing a single ($n = 1$) Pd nanoparticle used in our experiments. **g** Schematic depiction and zoomed-in DF image of the reaction zone of the empty catalyst bed without Pd nanoparticles ($n = 0$), used in our experiments as a negative control.

It is also visible that the counts vs. $\alpha_{CO}$ curve is slightly wider and flatter, and exhibits a significantly larger standard deviation (shaded area) for $n = 10$, in particular in the low reaction rate regimes at small and high $\alpha_{CO}$ values. Reducing temperature further amplifies this effect as the signal approaches the noise floor for $n = 10$, with counts in the 10–40 range at $T = 280\,°C$ (Fig. 2k, l). Consequently, it becomes increasingly difficult to extract the $CO_2$ signal, and we are approaching the limit of detection from a conventional data analysis perspective, where simply the number of counts is extracted from the QMS signal.

In principle, various strategies to address this issue exist, e.g., digital filtering[50], Fourier transforms[51,52], wavelet transforms[53], etc., or statistical methods, like principal component analysis (PCA)[54], among

others. When applied correctly, they enable the extraction of the desired signal from the accompanying noise and enhance the visibility of the underlying patterns in the data. However, these well-established denoising techniques have inherent limitations in the context of denoising QMS signals. Specifically, digital filtering can inadvertently remove crucial signals if they overlap with noise frequencies[55], Fourier and wavelet transforms may falter if the signal doesn't meet their assumptions about periodicity or localization[56] and PCA, which assumes the primary variance in data is due to the signal, might not be effective if noise has high variance or the signal is confined to a tight range[57]. Given these challenges, a denoising auto-encoder (DAE), i.e., an artificial neural network, offers a promising solution since it can be

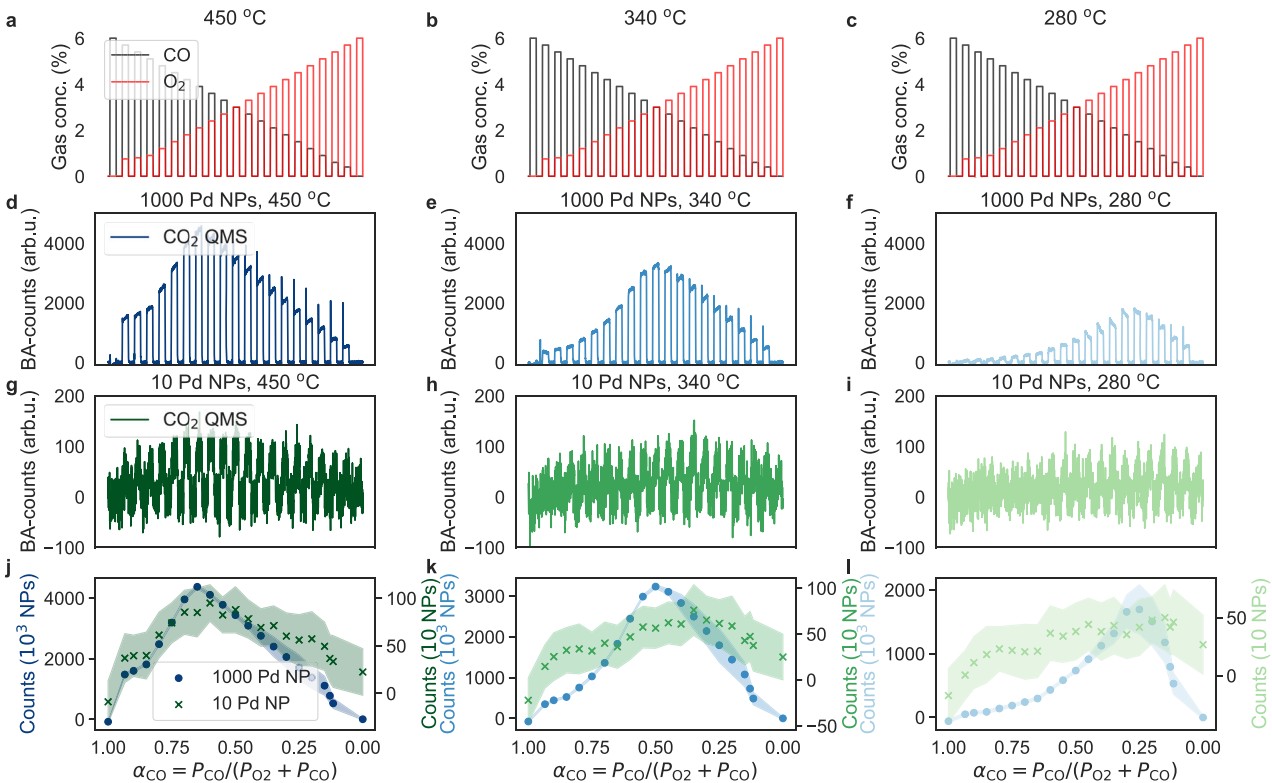

**Fig. 2 | CO oxidation experiments at different temperatures on 1000 and 10 Pd nanoparticles.** CO and $O_2$ pulse sequence applied at a reactor temperature of (**a**) $T = 450\,°C$, (**b**) $T = 340\,°C$, (**c**) $T = 280\,°C$. Note that we start all experiments with a pure 6% CO pulse in Ar carrier gas and subsequently decrease the relative CO concentration and increase the relative $O_2$ concentration, such that $0 \leq \alpha_{CO} = P_{CO}/(P_{O_2} + P_{CO}) \leq 1$, while keeping the total reactant concentration constant at 6%. Corresponding baseline-adjusted (BA - see Methods section on pre-processing for explanation of the BA-procedure) $CO_2$ counts measured by the QMS for 1000 Pd nanoparticles at (**d**) $T = 450\,°C$, (**e**) $T = 340\,°C$ and (**f**) $T = 280\,°C$. We note distinct responses at all three temperatures with the expected decrease in reaction rate for lower $T$. The "transient" overshoots observed for small $\alpha_{CO}$ values are the consequence of the mass flow controller "dial-in" to the correct flow rate. (**g**–**i**) Corresponding $CO_2$ counts measured by the QMS for the 10 Pd nanoparticle sample. While at $T = 450\,°C$ a relatively clear $CO_2$ signal is still obtained, it becomes

increasingly weaker at the lower temperatures. **j** Mean value of the measured $CO_2$ QMS counts for each reactant pulse plotted vs. the corresponding $\alpha_{CO}$ value for both the $n = 1000$ and the $n = 10$ Pd nanoparticle samples and obtained at $T = 450\,°C$. Shaded area depicts the standard deviation of measured BA-counts across each respective pulse. We note a maximum in the $CO_2$ formation rate at $\alpha_{CO}^{max} = 0.65$ for both $n = 1000$ and 10, as well as a significantly larger deviation of the derived counts for $n = 10$. **k** Same as (**j**) but for $T = 340\,°C$. We note a shift of $\alpha_{CO}^{max}$ to lower values for both samples due to increasing CO poisoning and a broadening/flattening of the overall trend for $n = 10$, which is the consequence of the system approaching the detection limit of the QMS and the consequent significant uncertainty in the derived counts. **l** Same as (**j**) and (**k**) but for $T = 280\,°C$. We note an even further reduction of $\alpha_{CO}^{max}$ and further uncertainty increase for $n = 10$. Source data are provided as a Source Data file.

trained to discern the structure of complex noisy data and reconstruct the underlying signal[58] (see Supplementary Note 3 and Supplementary Table 1). In practice, this is achieved by introducing examples of simulated signals with experimentally measured noise and training the network to reconstruct the (original) simulated signal. By also constraining the autoencoder's latent space to have a step function distribution, we introduce a prior, which enables denoising well below the signal-to-noise ratio (SNR) of 1 and ensures robustness in the final representations[59].

**A constrained DAE to improve the QMS limit of detection**
We applied the DAE to our QMS experiments by using experimentally measured QMS readout consisting of intrinsic measurement noise and noise induced by the nanoreactor setup (e.g., fluctuations in $CO_2$ concentration stemming from impurities in the used gases, small fluctuations of the mass flow controller and/or tiny leaks), combined with an underlying true $CO_2$ signal stemming from the catalytic reaction on the Pd nanoparticle(s), as input (Fig. 3). This combined signal is then compressed through encoder $E_\theta$ to form a latent space inside the bottleneck, which is constrained through a consistency loss to form a step function distribution. This compressed representation is then upsampled in decoder $D_\theta$ to form the reconstructed true signal, i.e.,

the QMS readout with a deconvolved underlying true $CO_2$ signal. In this design, the DAE can learn complex non-linear relationships between the noise and the true signal, enabling it to handle the kind of noise-signal interactions that might confound the other noise-reducing methods mentioned above. For a comparison between the DAE and other denoising techniques applied to our QMS data, see Supplementary Note 4 and Supplementary Figs. 7–9.

To evaluate the performance of the DAE, we compare the measured baseline-adjusted (BA) QMS $CO_2$ counts (see Methods section on preprocessing for explanation of the BA-procedure) as function of $\alpha_{CO}$ for $n = 1000$ and 10, for temperatures between 280 and 450 °C, as well as the DAE-denoised $CO_2$ counts for $n = 10$ (Fig. 4). Focusing first on $n = 1000$ at 450 °C, the catalyst exhibits the highest reaction rate at $\alpha_{CO}^{max} = 0.65$ and all $CO_2$ pulses are clearly discernible (Fig. 4a). Executing identical analysis for $n = 10$ at 450 °C reveals individual $CO_2$ pulses and $\alpha_{CO}^{max} \approx 0.6$ (Fig. 4b). Passing the $n = 10$ QMS data obtained at 450 °C through the DAE reveals that it delivers more distinct and systematically in/decreasing $CO_2$ pulses, with a maximum rate at $\alpha_{CO}^{max} = 0.65$, in excellent agreement with $n = 1000$ (Fig. 4c). Repeating the same analysis for 280 °C reveals the anticipated reduction of the overall rate for both samples and a distinct shift of $\alpha_{CO}^{max}$ to $\alpha_{CO}^{max} \approx 0.2$, due to severe CO poisoning (Fig. 4d–f). Importantly, for $n = 10$, it is now

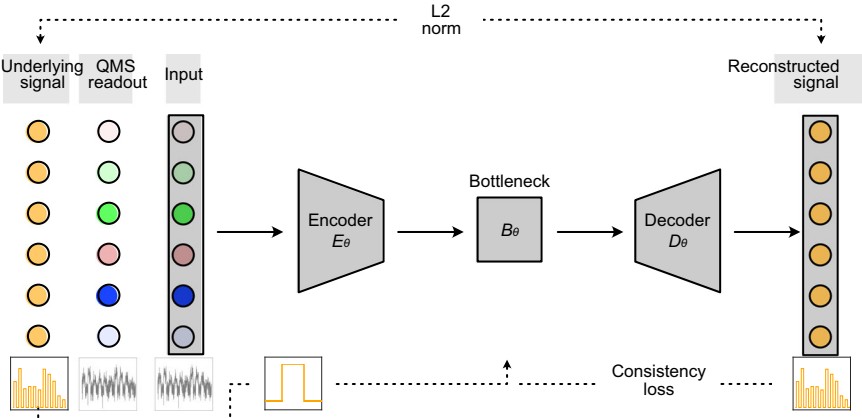

**Fig. 3 | The constrained denoising auto-encoder.** The underlying signal that we want to access in our experiments, i.e., the $CO_2$ production rate step functions as a function of time, is corrupted by both correlated and uncorrelated noise in the experimentally measured readout from the QMS. Therefore, the experimental QMS readout is input to a denoising autoencoder, which transforms the input into a minimally dimensioned representation of the underlying signal and then reconstructs it sans corruption, which means, it reconstructs the original signal, rather then the QMS readout that also contains different types of noise. The DAE is trained with a standard L2 norm between reconstructed and underlying signal, and the latent space is constrained by an L1 norm and a consistency loss, which forces the latent space to take on the form of the underlying signal.

clear that the standard analysis is approaching the limit of detection where the $CO_2$ pulses are stochastic, whereas the DAE-denoised data still exhibits the same clear trend as seen for $n = 1000$ with the standard analysis.

As the next step, we extract the mean $CO_2$ counts for each $\alpha_{CO}$ pulse along the $\alpha_{CO}$ sweep for $n = 1000$, obtained for temperature steps of 20 °C from 280 to 450 °C (Fig. 4g). This reveals a constant and stoichiometric $\alpha_{CO}^{max} = 0.65$ down to 380 °C, below which $\alpha_{CO}^{max}$ systematically shifts to smaller values due to increasing CO poisoning, and reaches $\alpha_{CO}^{max} = 0.25$ at 280 °C. Furthermore, it reveals a decrease in reaction rate from 5240 $CO_2$ counts at $\alpha_{CO}^{max}$ at 450 °C, to 1700 $CO_2$ counts at $\alpha_{CO}^{max}$ at 280 °C. Plotting the same data for $n = 10$ obtained for 10 °C temperature steps, and comparing the standard analysis (Fig. 4h) with the DAE-based analysis (Fig. 4i) reveals qualitatively very similar behavior. However, crucially, SNR is very poor for the standard analysis to the point where the underlying chemical dynamics become obscured by stochastic shifts in, e.g., $\alpha_{CO}^{max}$. This is different for the DAE-based data, which reproduces the $T$-dependent $\alpha_{CO}^{max}$ trend found for the $n = 1000$ sample very clearly (Fig. 4j–l). The situation is similar when comparing the absolute BA · $CO_2$ counts at $\alpha_{CO}^{max}$ at 450 °C. For $n = 1000$, we count 5240 $CO_2$ molecules, whereas we count 90 and 65 for $n = 10$ for the standard and DAE-based analysis, respectively. The DAE-based value of 65 is indeed proportional to a $100\times$ decrease in catalytic surface area (compared to $n = 1000$), within a relative error of roughly 10%. Thus, the measured raw value of 90 counts likely overestimates the true $CO_2$ production rate due to residual noise or background signal, highlighting the DAE's ability to accurately correct for that. The results demonstrate that the DAE effectively reconstructs the catalysts' $CO_2$ production rate and, e.g., the temperature-dependent poisoning state of the system, even when dealing with considerably noisy data and the QMS operated very close to its limit of detection.

### Resolving reaction products formed on Pd NPs with a DAE

We put the DAE to the test by further reducing the amount of catalyst by a factor 10, down to a single Pd nanoparticle with a surface area of $\approx 7200 \text{ nm}^2$. We used a chip with $n = 1$ (cf. Fig. 1f) and an empty chip with $n = 0$ as the control (cf. Fig. 1g). Clearly, even at the highest temperature of 450 °C, the $\alpha_{CO}$ sweeps for $n = 0$ and 1 look very similar using the standard analysis (Fig. 5a, b). They are characterized by small stochastic $CO_2$ pulses, indicating a certain level of background $CO_2$ in the reactant gas mixture, as well as the signal stemming from the $CO_2$

produced by the catalyst particle drowning in background noise. This is further corroborated by extracting the BA-$CO_2$ counts and plotting them vs. $\alpha_{CO}$ for both $n = 1$ (7 identical sweeps) and 0 (5 identical sweeps), which reveals that no clear activity trend as a function of $\alpha_{CO}$ is resolved for $n = 1$, despite a generally slightly higher number of counts compared to $n = 0$ (Fig. 5c and Supplementary Figs. 6, 10).

Applying the DAE first to the $\alpha_{CO}$ sweep at 450 °C for $n = 1$ reveals a different picture (Fig. 5d). Rather than stochastic $CO_2$ pulses, a clear trend of in- and decreasing $CO_2$ counts along the $\alpha_{CO}$ sweep is recovered, which again exhibits a maximum at the stoichiometric $\alpha_{CO}^{max} = 0.65$. This is in very good agreement with the corresponding experiments for $n = 10$ and 1000. Applying the DAE to the $\alpha_{CO}$ sweep at 450 °C for the $n = 0$ control outputs a close to completely flat baseline at zero BA-counts, corroborating that the DAE is able to eliminate the noise contaminating the raw QMS signal (Fig. 5e).

Encouraged, we apply the DAE to $\alpha_{CO}$ sweeps measured at temperatures 330 to 450 °C for $n = 1$ and plot the extracted $CO_2$ counts vs. $\alpha_{CO}$ (Fig. 5f). Remarkably, we clearly resolve the anticipated increase of the reaction rate to an $\alpha_{CO}^{max} = 0.65$ for decreasing $\alpha_{CO}$, and the subsequent decrease of the reaction rate upon decreasing $\alpha_{CO}$. Furthermore, the extracted number of $CO_2$ counts $\approx 6$ is $\approx 10$ times lower than for $n = 10$ at 45 °C, which is in excellent agreement with a reduction of the active surface area by a factor of 10 from $n = 10$ to 1. Finally, upon reduction of temperature, we see an indication of a shift of $\alpha_{CO}^{max}$ to lower values at 410 °C, before the DAE-signal also drops below the limit of detection of a single count. Similar analysis of the $n = 0$ control at 450 °C consistently delivers $CO_2$ counts very close to or significantly below the limit of detection of a single count and thus confirms the ability of the DAE to resolve $CO_2$ produced by the single Pd nanoparticle (Fig. 5f). As a last point, we note that the low absolute counts observed for single nanoparticles (c.f. Pd $n = 1$ in Fig. 5f) raise the question of potential error in the activity analysis in this regime of very few counts. While it is true that low count numbers can, in principle, lead to larger relative uncertainties, the DAE's denoising capabilities mitigate this issue. Specifically, the DAE is trained to distinguish genuine signals from noise, even when the signal is very weak. Therefore, while the absolute number of counts might be low, the proportion of those counts representing actual $CO_2$ production is very high after DAE processing compared to the raw data and thus the counts predicted by the DAE are indeed significant. Furthermore, the consistency of the DAE-derived trends across multiple independent measurements (c.f.

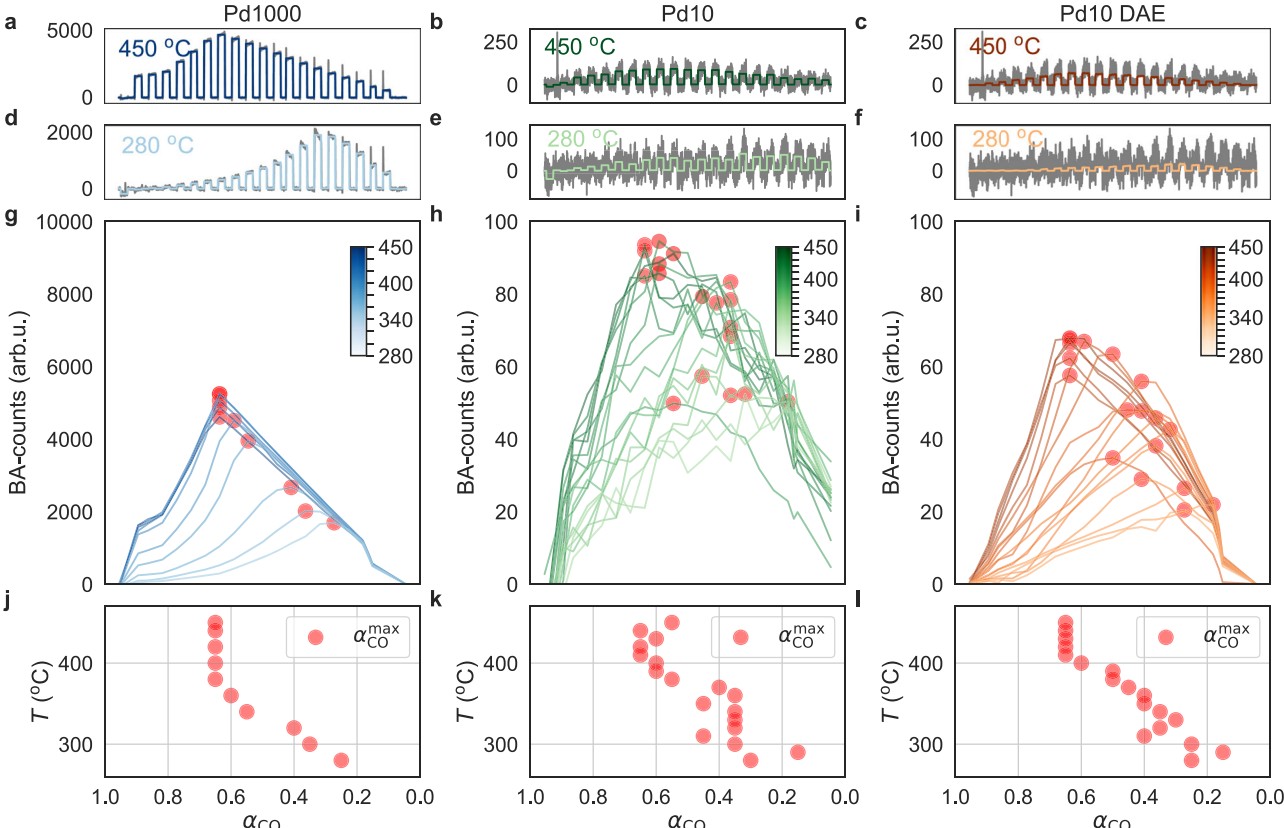

**Fig. 4 | Direct comparison of standard and DAE-enhanced QMS readout for 1000 and 10 Pd nanoparticles. a** Baseline-adjusted (BA - see Methods section on preprocessing for explanation) raw QMS counts for $CO_2$ (gray line) together with the mean $CO_2$ signal (blue line - obtained as the average measured BA-count across each pulse) for $n = 1000$ Pd nanoparticles across the entire $\alpha_{CO}$ range, $\alpha_{CO} \in (0, 1)$, and at 450 °C. **b** BA-QMS counts for $CO_2$ (gray line) together with the mean $CO_2$ signal (green line) for $n = 10$ Pd nanoparticles across the entire $\alpha_{CO}$ range and at 450 °C. **c** BA-QMS counts for $CO_2$ (gray line) together with the $CO_2$ signal denoised by the DAE (orange line) for $n = 10$ across the entire $\alpha_{CO}$ range, $\alpha_{CO} \in (0, 1)$, and at 450 °C. (**d**–**f**) Same as (**a**–**c**) but at 280 °C. **g** BA-mean $CO_2$ counts for $\alpha_{CO}$ sweeps at reactor temperatures ranging from 280 to 450 °C in 20 °C steps for $n = 1000$. Red dots indicate the $\alpha_{CO}$ value corresponding to the highest reaction rate, $\alpha_{CO}^{max}$.

**h** Same as (**g**) but for $n = 10$, using 10 °C temperature steps. **i** Same as (**h**) but with the QMS signal denoised by the DAE. **j** $\alpha_{CO}^{max}$ values extracted from (**g**) for all temperatures for $n = 1000$, plotted as a function of $\alpha_{CO}$. We note the constant stoichiometric $\alpha_{CO}^{max}$ value of 0.65 down to $T = 360$ °C. Below this temperature, we find a systematic shift to lower $\alpha_{CO}^{max}$ values, as a consequence of increasing CO poisoning. **k** Same as (**j**) but for $n = 10$. We note the qualitatively similar trend compared to $n = 1000$, but also the significantly higher spread in the data points. **l** Same as (**k**) but based on the DAE-denoised QMS signal in (**i**). Clearly, the uncertainty in $\alpha_{CO}^{max}$ is significantly reduced and the $T$-dependent trend of $\alpha_{CO}^{max}$ for both $n = 1000$ and 10 is now very similar. The small discrepancies are discussed in the text. Source data are provided as a Source Data file.

Fig. 5) corroborates the reliability of the activity trends obtained by the DAE, even for single nanoparticles.

We put these results into perspective by comparing them to the data obtained for $n = 10$ and 1000. Specifically, we plot the $\alpha_{CO}^{max}$ values for $n = 1000$, 10 DAE-denoised and 1 DAE-denoised for 400 °C $\leq T \leq$ 450 °C, i.e., the range for which the $n = 1$ DAE-signal is above the limit of detection (Fig. 5g). These values are very similar for all three samples and very close to the stoichiometric value of $\alpha_{CO}^{max} = 0.66$. This is important because it confirms that the catalyst experiences identical reaction conditions in all three implementations in terms of the number of particles. We also compare the $CO_2$ counts obtained at $\alpha_{CO}^{max}$ for 400 °C $\leq T \leq$ 450 °C normalized by the number of particles on the respective sample, i.e., $n = 1000$, 10 and 1 (Fig. 5h). This reveals particle-number-normalized $CO_2$ count values between 4 and 6 and thus corroborates direct scaling of extracted QMS counts with catalyst surface area, within the expected level of uncertainty that is imposed, e.g., by slightly different particle dimensions on each of the three chips.

To ensure that the $CO_2$ signal observed in the catalytic experiments originates exclusively from surface reactions on Pd nanoparticles, we conducted a control experiment using an identical nanofluidic chip lacking any catalytically active material. Pulses of $CO_2$ diluted in Ar were flushed through the chip under identical conditions

(450 °C, 2 bar, 20 mL min$^{-1}$ total flow), and the resulting QMS signal was analyzed. As detailed in Supplementary Fig. 17a, b, no evidence of spurious $CO_2$ formation or retention effects was observed. Furthermore, the DAE successfully retrieved the shape and timing of the low-intensity $CO_2$ pulses even when the raw signal approached the noise floor, confirming the robustness of the signal processing method and excluding non-catalytic sources of $CO_2$ response.

## DAE for ethylene hydrogenation with a less sensitive QMS

To demonstrate the versatility of the DAE presented in this work, as well as to test it from a different perspective and using a different catalytic reaction, we investigated ethylene hydrogenation on 1000 Pd nanoparticles using a QMS with lower intrinsic sensitivity. This experiment was designed to illustrate that our system is not limited to CO oxidation and not only can reduce the amount of catalyst surface area required for online QMS measurements but also can be used to increase the sensitivity of lower grade QMS instrument. Therefore, we exchanged the high-sensitivity Hiden HAL/3F PIC QMS used in the CO oxidation experiments by a significantly less sensitive Pfeiffer Prisma QME200. Furthermore, as in the CO oxidation experiments, we used a pulsed gas sequence comprised of alternating pulses of pure Ar and $C_2H_4$ mixed with $H_2$ in different ratios in Ar at a total constant reactant

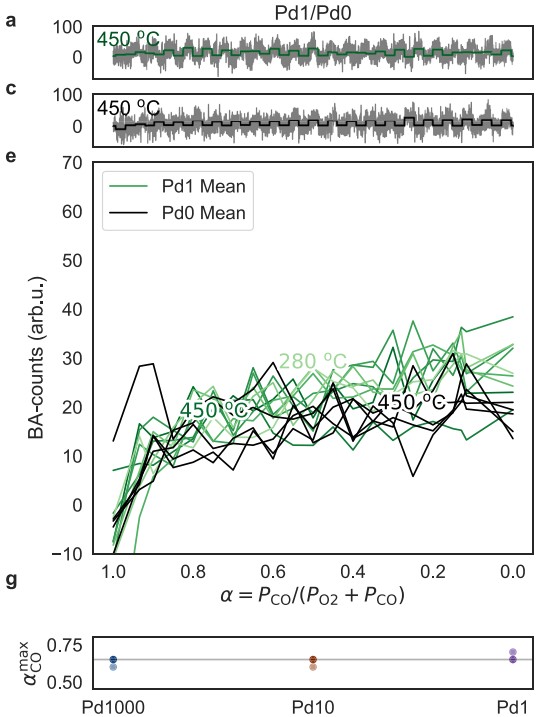

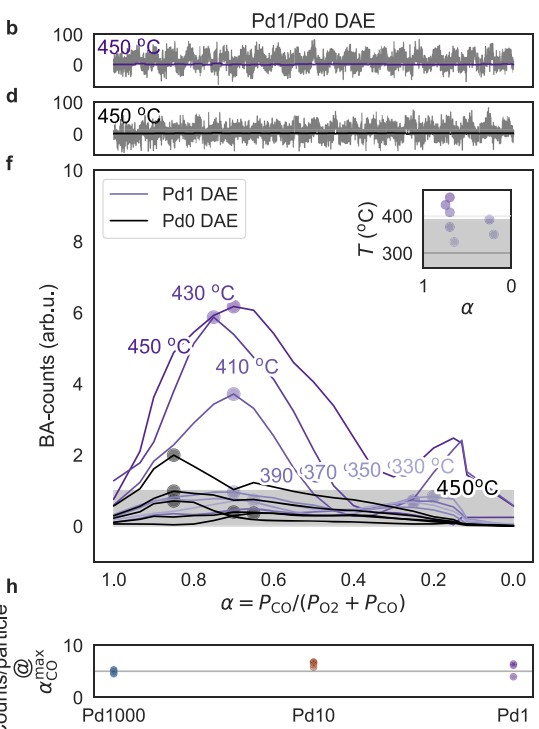

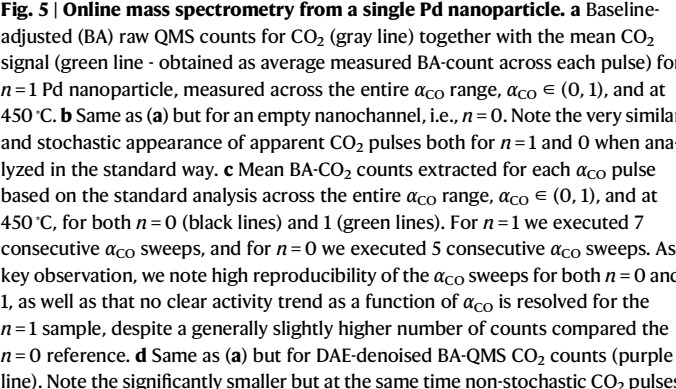

**Fig. 5 | Online mass spectrometry from a single Pd nanoparticle. a** Baseline-adjusted (BA) raw QMS counts for $CO_2$ (gray line) together with the mean $CO_2$ signal (green line - obtained as average measured BA-count across each pulse) for $n = 1$ Pd nanoparticle, measured across the entire $\alpha_{CO}$ range, $\alpha_{CO} \in (0, 1)$, and at 450 °C. **b** Same as (**a**) but for an empty nanochannel, i.e., $n = 0$. Note the very similar and stochastic appearance of apparent $CO_2$ pulses both for $n = 1$ and 0 when analyzed in the standard way. **c** Mean BA-$CO_2$ counts extracted for each $\alpha_{CO}$ pulse based on the standard analysis across the entire $\alpha_{CO}$ range, $\alpha_{CO} \in (0, 1)$, and at 450 °C, for both $n = 0$ (black lines) and 1 (green lines). For $n = 1$ we executed 7 consecutive $\alpha_{CO}$ sweeps, and for $n = 0$ we executed 5 consecutive $\alpha_{CO}$ sweeps. As key observation, we note high reproducibility of the $\alpha_{CO}$ sweeps for both $n = 0$ and 1, as well as that no clear activity trend as a function of $\alpha_{CO}$ is resolved for the $n = 1$ sample, despite a generally slightly higher number of counts compared the $n = 0$ reference. **d** Same as (**a**) but for DAE-denoised BA-QMS $CO_2$ counts (purple line). Note the significantly smaller but at the same time non-stochastic $CO_2$ pulses

resolved by the DAE compared to the standard analysis. **e** Same as (**d**) but for an empty nanochannel, i.e., $n = 0$. Note that using the DAE, a flat baseline at zero BA-counts is obtained for the empty nanochannel. **f** Same as (**c**) but for DAE-denoised data, where distinct reaction rate maxima are resolved for $n = 1$ (purple curves), with $\alpha_{CO}^{max}$ ranging between 0.65 and 0.6 for temperatures between 450 and 410 °C (inset), in good agreement with the earlier results for $n = 10$ and 1000. For lower temperatures, as well as for $n = 0$ (black lines representing multiple sweeps at 450 °C), the DAE outputs counts < 1, which is physically unreasonable and thus defined as the limit of detection. **g** $\alpha_{CO}^{max}$ values for $n = 1000$ (blue), 10 DAE-denoised (orange) and 1 DAE-denoised (purple) for 400 °C $\leq T \leq$ 450 °C, i.e., the range for which the $n = 1$ DAE-signal is above the limit of detection. **h** $CO_2$ counts normalized by the number of particles on the respective sample, obtained at $\alpha_{CO}^{max}$ for $n = 1000$ (blue), $n = 10$ DAE-denoised (orange) and 1 DAE-denoised (purple) for 400 °C $\leq T \leq$ 450 °C. Source data are provided as a Source Data file.

concentration of 1% and 4 bar inlet pressure (Fig. 6a). This sequence was designed to transition from pure $C_2H_4$ to pure $H_2$, and back to pure $C_2H_4$, while monitoring the formed ethane ($C_2H_6$) with the QMS. The experiment was conducted at temperatures of 85, 90, 120, 160, and 170 °C, respectively. Analyzing the raw normalized $C_2H_6$ ion current measured by the QMS alongside the filtered signal derived using the DAE for three representative temperatures reveals a distinct QMS response at the highest temperature of 170 °C that quickly drowns in noise when the reaction temperature is reduced (Fig. 6b, see Supplementary Fig. 16 for all measured temperatures). Accordingly, it again showcases the ability of the DAE to extract the weak QMS signals from noise in this regime, as the DAE predicted $C_2H_6$ QMS response exhibits clear pulses even at the lowest temperature of 85 °C.

To further analyze the obtained results, and in analogy to the CO oxidation experiments, we plot the averaged ion counts for each step of the pulse sequence predicted by the DAE as a function of the mixing parameter $\alpha_{C_2H_4} = P_{C_2H_4}/(P_{C_2H_4} + P_{H_2})$ for the down-sweep (Fig. 6c) and up-sweep (Fig. 6d), respectively. In the down-sweep, as the concentration of $C_2H_6$ decreases, the $C_2H_6$ production initially peaks and then decreases, indicative of first-order kinetics with respect to $C_2H_4$, followed by a regime of negative-order kinetics. Conversely, in the up-sweep, the system exhibits first-order kinetics relative to $H_2$, followed by a zero-order plateau, and then transitions into negative-order

kinetics. This behavior is attributed primarily to the competitive adsorption between ethylene and hydrogen on the Pd catalyst surface. During the up-sweep, H atoms accumulate more rapidly than ethylene can compete for adsorption sites, resulting in a surface saturated with H and thus a zero-order kinetic regime with respect to $H_2$[60,61]. In contrast, starting with a surface saturated with $C_2H_4$ during the down-sweep hinders H adsorption once introduced, preventing the establishment of the zero-order regime[62]. These findings align with the Horiuti-Polanyi mechanism, where surface coverage and competitive adsorption play crucial roles[63,64]. Overall, these results demonstrate that the DAE in combination with nanofluidic reactors enables the extraction of detailed kinetic information also from a much slower hydrogenation reaction, and that at lower overall reactant concentration and with a less sensitive QMS than for the CO oxidation experiments above, using a small catalyst bed of 1000 Pd nanoparticles.

## Discussion

We have demonstrated how the combination of nanofluidic reactors and DAE-based deep learning enables the reduction of catalyst surface area required for online QMS analysis of reaction product in the gas phase by $\approx$ 3 orders from the current SotA, down to the level of single nanoparticles. This breakthrough was enabled by the ability of nanofluidic reactors to focus reaction products from tiny catalyst surfaces

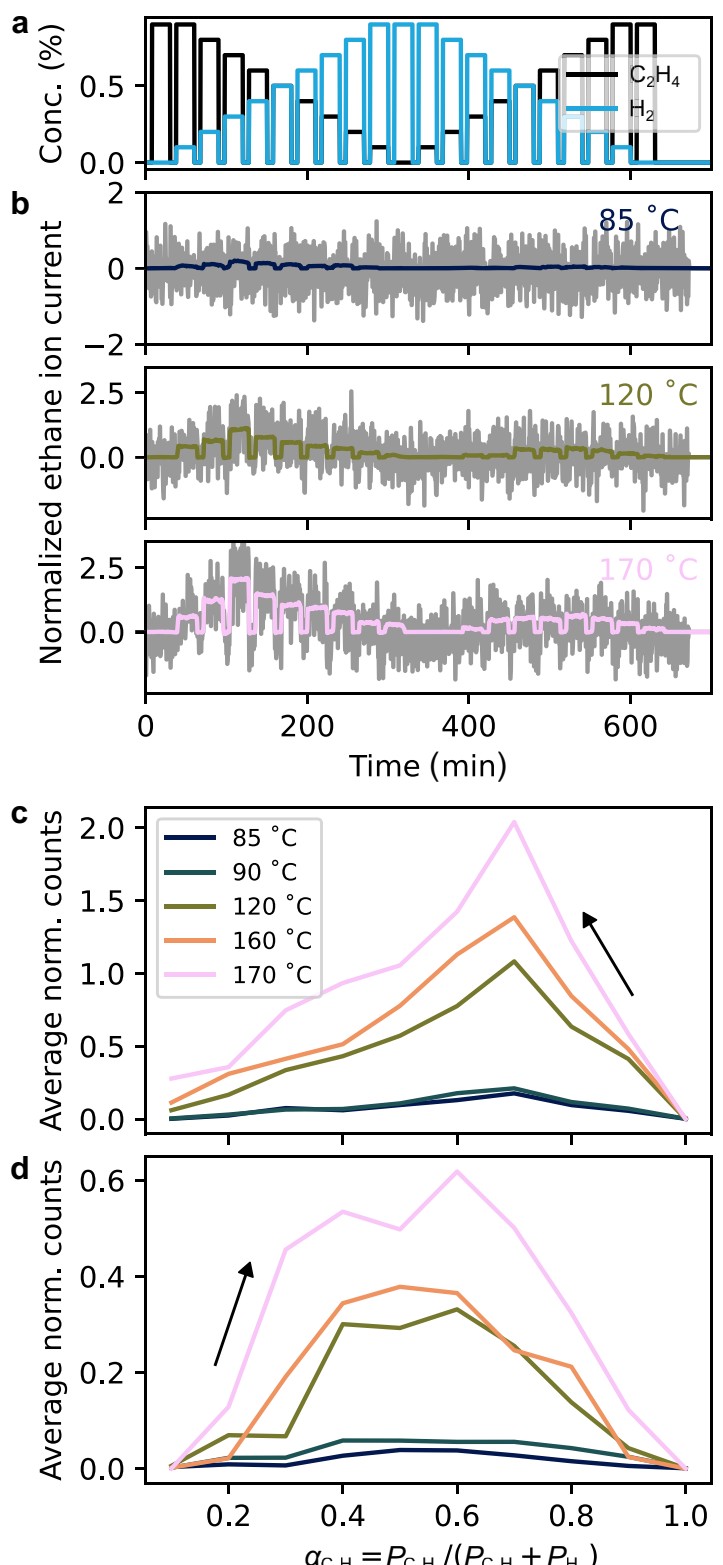

**Fig. 6 | Ethylene hydrogenation experiments at different temperatures on 1000 Pd nanoparticles using a lower sensitivity QMS. a** Mixed $C_2H_4$ and $H_2$ pulses with systematically varied concentrations in Ar carrier gas applied to the 1000 Pd nanoparticle catalyst bed at different temperatures. **b** Baseline-adjusted (BA - see Methods section on preprocessing for explanation) normalized $C_2H_6$ reaction product counts measured by QMS at $m/z = 30$ (gray lines), together with the signal denoised by the DAE (colored lines) for 85, 120, and 170 °C catalyst bed temperatures. The colors refer to the corresponding temperatures shown in panel (**c**). **c**, **d** BA-adjusted $C_2H_6$ counts as a function of the reactant mixing parameter $\alpha_{C_2H_4} = P_{C_2H_4}/(P_{C_2H_4} + P_{H_2})$, calculated for every step of the pulse sequence shown in (**a**). The arrows indicate the direction of the sequence: in (**c**) $\alpha_{C_2H_4}$ goes from 1 to 0 in steps of 0.1, in (**d**) $\alpha_{C_2H_4}$ goes from 0 to 1 in steps of 0.1. The colors in (**d**) match the temperatures reported in (**c**). Source data are provided as a Source Data file.

towards analysis, and the high capacity of autoencoders to discern weak signals from noise. We have illustrated this on the example of the CO oxidation reaction on Pd nanoparticle model catalysts $A \approx (0.0072 \pm 0.00086)\,\mu m^2 = (7200 \pm 860)\,nm^2$ per particle, localized inside well-mixed nanofluidic catalyst beds. We analyzed nanoreactors with $n = 1000, 10, 1$ and $0$ Pd particles across a temperature range from 280 to 450 °C and found the characteristic dependence of the reaction rate on $\alpha_{CO} = P_{CO}/(P_{O_2} + P_{CO})$, as well as a stoichiometric highest reaction rate at $\alpha_{CO}^{max} = 0.66$ for the highest temperatures, with a distinct shift to lower $\alpha_{CO}$ at lower temperatures, due to CO poisoning. While standard data analysis enabled meaningful results for $n = 1000$ and 10 (at the highest temperatures) only, DAE denoising produced reliable QMS data even for $n = 1$, and facilitated the analysis of the reaction dynamics in terms of $\alpha_{CO}$-dependent surface poisoning also for a single Pd nanoparticle.

In a wider perspective, we highlight that our approach is not specific to CO oxidation, and we have shown that it can be adapted for other catalytic reactions, such as the hydrogenation of $C_2H_4$ to $C_2H_6$ on Pd, and in principle any measurement problem, in which the structure of the underlying true signal is known. Our results thus constitute a new paradigm to significantly improve the resolution in online reaction analysis in (single-particle) catalysis and advocate deep learning to extract tiny QMS signals from noise, as well as to boost the sensitivity of QMS instruments beyond the limitations imposed by the used hardware.

## Methods

### Deep learning architecture

We employed a constrained denoising autoencoder to denoise the experimental QMS data. This architecture was designed to accept an input shape of 6400 timesteps, equaling the full size of a single $\alpha_{CO}$ sweep. The encoder segment consists of seven convolutional layers, each with 32 neurons and a kernel size of 9. These layers are systematically interspersed with max-pooling operations to achieve optimal data compression and utilize the leaky rectified linear unit (LeakyReLU) as its activation function due to its proven efficacy in handling complex non-linearities whilst avoiding mode collapse, vanishing and exploding gradients. Following this sequence, the encoded data is transformed into its latent space through a dense layer, reducing its dimensionality to equal the number of steps of the $\alpha_{CO}$ sweep. This compressed representation is then reshaped and combined with the previously encoded data to retain both quantitative and structural information of the underlying signal to the decoder.

In the symmetrical decoder segment, the model reconstructs the data through another series of seven convolutional layers, each followed by an upsampling operation, ensuring accurate data restoration. The final layer then produces a reconstruction of the initial data.

To enhance the model's accuracy and ensure it only outputs functions representative of the underlying signal, we incorporated a consistency loss function which compares mean values across specific data segments of the reconstructed input with their corresponding latent space representations. This function ensures that the autoencoder's reconstructed output aligns structurally with the original data. The network is trained with an MSE function between target and predicted data segments, as elaborated upon in Supplementary Table 1 and corresponding discussion. Overall, this architecture effectively balances noise reduction, feature retention, and data consistency.

The architecture was implemented through TensorFlow[65] and the code can be found in the corresponding GitHub page.

### Synthetic data generation

The system utilizes two generator functions: a generator for the underlying signal and a generator for the noise. The noise is generated first, and is defined by its standard deviation and shape of the gas pulses. It is generated in two parts; firstly from typical stochastic fluctuations in the QMS readout itself such as thermal noise, environmental fluctuations, residual particles (flicker noise), shot noise and ion feedback noise. We model these by a Gaussian distribution with a high range of variability, to encapsulate the (white noise) effects of all these sources in the training set. Secondly, there is contamination noise, resultant of reagent reactions which occur outside the catalytically active surface area of the nanoparticles. We model these by an added term for each gas pulse, proportional to the concentration of input reagents to the system, also with a high variability.

The signal generator creates a step function with a desired signal-to-noise ratio $SNR = \frac{\mu_{signal}}{\sigma_{noise}}$, which is chosen during training to match the SNR of 1-10 Pd NP reactor readouts based on the given noise. The steps are always placed the same (i.e., in an on-off pulsing form), but each with a random signal value. Thus, there is no inherent inductive bias regarding the relation of one pulse to another within the same overall step function.

### Deep learning training

The training consists of two steps. Firstly, a curriculum learning scheme is employed on artificially generated signals on Gaussian-distributed noise with a pre-defined SNR. The SNR is drawn from a uniform distribution in the range $SNR \in (0.95, 1)$, quantitatively corresponding roughly to the highest amount of experimentally measured catalytic activity of the $n = 10$ Pd sample at $T = 450$ °C. The lower end of this range decreases by a factor of 0.05 to a minimum of 0 each time the network is trained to convergence, as defined by an early stopping mechanism with a patience of 32 epochs. This scheme helps prevent problems of vanishing gradient and thus mode collapse result. of directly training on extremely-low SNR examples. The artificial data generation allows the model to train on a broad range of noises and signal distributions, increasing its robustness by forcing it to learn generalizable functions for denoising. The loss curve for this training can be found in Supplementary Fig. 14.

The second step consists of training on artificial signal distributions in the range $SNR \in (0, 1)$, corresponding to the full range of SNRs considered in this work, generated on examples of experimentally measured noise. This fine-tuning step ensures that the final functions learnt by the network work appropriately for the particular noise distributions relevant to the experimental conditions at inference time. The noise is here defined as the output of a CO oxidation sequence of a $n = 0$ Pd sample chip at $T = 40$ °C.

### Preprocessing

QMS data was preprocessed using a custom Python script (see Code Availability) developed for this study. To adjust for baseline shifts of the QMS signal over time, which can occur if the pressure inside the UHV chamber on which the QMS is mounted has not yet entirely stabilized, the following procedure is applied. First, a separate linear fit is made to the QMS signal for each pulse in pure Ar. Secondly, this fit is subtracted from said QMS signal and from the QMS signal of the next corresponding $\alpha_{CO}$ pulse in time. This shifts the baseline to a consistent level for each individual pulse within a sweep. This procedure is referred to as baseline adjustment (BA) in the main text. Outliers, defined as individual QMS data points deviating more than three standard deviations from the mean, were removed. The mean QMS count value for each period was thereafter calculated and thus constituted the standard analysis for this study. Finally, the data are scaled down by a factor 100 to bring it in range with the order of magnitude of a standard Gaussian, which is how the neural network's weights are initialized and therefore stabilizes training.

### Experimental setup

Experiments were performed on a setup reported in detail earlier[35,36] (see Supplementary Fig. 1). In brief, it is comprised of an elevated/ atmospheric pressure gas mixing/handling unit connected to the inlet

side of a nanofluidic chip holder that, on the outlet side, is connected to an ultrahigh vacuum system (base pressure of $10^{-10}$ mbar) equipped with a triple filter QMS equipped with a pulsed ion counting detector (Hiden HAL/3F PIC) and a gold-plated ion source. The QMS was operated in SEM mode and at 12 s sampling rate. In the case of ethylene hydrogenation, a Pfeiffer Prisma QME200 was used to detect the ethane formation. No absolute calibration was performed; instead, all signals were analyzed comparatively using consistent instrument settings, flow conditions, and background subtraction protocols for all executed experiments. The gas mixing/handling unit was built from quarter-inch stainless steel tubing with Vacuum Coupling Radiation (VCR) (Swagelok) fittings as connectors. The nanofluidic chip was hosted in a stainless steel connection block with welded VCR fittings, which provided gas and QMS connections. The connection block featured interior gas lines connected to exterior flexible stainless steel tubing. The gas inlet and outlet of the chip were sealed with perfluoroelastomer (FPM) O-rings to the connection block. To avoid unwanted background signals, a constant flow of Ar (18 mL min$^{-1}$) was flushed around the O-rings, limiting any diffusion to Ar. Connectors on the connector block allowed for integration with four Bronkhorst Low $\Delta P$ mass flow controllers and a pressure controller (max pressure of 10 bar). This system facilitates the creation of a gas mixture with up to four different gases and a defined inlet pressure up to 10 bar upstream of the mass flow controllers (here we used 4 bar). All gas flows and pressures were controlled and monitored through a customized LabVIEW program. The system can be heated up to 450 °C using a microfabricated resistive heater on the backside of the chip. Temperature readout was performed through a four-wire resistance temperature probe microfabricated on the backside of the chip at the position of the nanofluidic system. The chip was electrically connected for heating and temperature readout through six gold-plated electronic spring spins embedded in a machined ceramic block. The pins are connected to a temperature controller (Lakeshore 335) operated via a LabVIEW program. The chips are calibrated by placing one in an oil bath with a thermocouple readout to monitor the temperature before any measurements are performed, see Supplementary Fig. 15. The chip holder is equipped with a water-cooled copper block to maintain a constant temperature, helping to reduce mechanical movements due to heat transfer into the holder assembly.

### Sample mounting and pre-treatment
Prior to mounting of a nanofluidic chip, the pipe connecting the outlet of the nanofluidic chip to the inlet of the QMS vacuum chamber are heated to 353 K to minimize water adsorption. Once the chip was mounted, the system was pumped for up to 72 h, until the base pressure of the QMS chamber reached the desired of $10^{-10}$ mbar. After mounting and pumping/baking, the sample was heated to 280 °C and subsequently exposed to 20 cycles of alternating pulses of 10% CO in Ar carrier gas and 15% O$_2$ in Ar carrier gas, each 15 min long, followed by a complete $\alpha_{CO}$ sweep, in order to activate the catalyst and reach a stable QMS signal baseline.

### CO oxidation experiments
For the CO oxidation $\alpha_{CO}$ sweep experiments, CO (10% in Ar) and O$_2$ (15% in Ar) were used with Ar carrier gas (99.99999% purity). The inlet pressure was set to 4 bar, and a total flow of 10 mL min$^{-1}$ through the microchannels was applied. The experiment sequence consists of 15 min pulses of CO/O$_2$ mixtures at a constant 6% percent reactant concentration, separated by 15 min in pure Ar (Fig. 2a–c) at constant temperatures ranging from 280 to 450 °C. In each pulse, the concentration of CO/O$_2$ is varied such that $\alpha_{CO}$ is systematically varied from 1 to 0 in steps of 0.05 for each subsequent pulse. The only exception is the first (and last) two pulses, which vary in steps of 0.11 (0.02) and 0.02 (0.11), respectively. This is due to instrumental limitations of the MFC controllers at very low values of absolute flow,

where they are unable to control the flow accurately. Each experiment is also started with a single 30 min pulse of 4% O$_2$ and 2% CO, followed by a 15 min Ar pulse, to reset the state of the catalyst before each sequence.

### C$_2$H$_4$ hydrogenation experiments
The ethylene hydrogenation experiments were conducted with C$_2$H$_4$ (2% in Ar) and H$_2$ (3.5% in Ar). The mixture of gases were flown through the microchannels with a total flow rate of 20 mL min$^{-1}$ and an inlet pressure of 4 bar. The sequence of pulses consisted in 45 min of exposure to C$_2$H$_4$/H$_2$ mixtures at constant 1% reactant concentration, followed by 15 min of pure Ar (see Fig. 6a). The concentrations were chosen such that the mixing parameter $\alpha_{C_2H_4}$ was varied from 1 to 0 and from 0 to 1 in steps of 0.1. The experiments were conducted while heating the nanofluidic chip to the constant temperatures of 85, 90, 120, 160 and 170 °C.

### Nanofabrication of the nanoreactor chip
Fabrication of the nanofluidic systems was carried out in the cleanroom facilities of Fed. Std.209 E Class 10 - 100, using electron-beam lithography (JBX-9300FS / JEOL Ltd), direct-laser lithography (Heidelberg Instruments DWL 2000), photolithography (MA 6 / Suss MicroTec), reactive-ion etching (Plasmalab 100 ICP180 / Oxford Plasma Technology and STS ICP), electron-beam evaporation (PVD 225 / Lesker), magnetron sputtering (MS150 / FHR), deep reactive-ion etching (STS ICP / STS) and wet oxidation (wet oxidation / Centrotherm), fusion bonding (AWF 12/65 / Lenton), and dicing (DAD3350 / Disco). In particular, the fabrication steps comprised the following processing steps of a 4-inch silicon (p-type) wafer:

Fabrication of alignment marks: spin coating HMDS (hexamethyldisilazane) adhesion promoter (MicroChem) at 3000 rpm for 30 s and soft baking on a hotplate at 115 °C for 120 s, spin coating UV5 (MicroChem) at 2000 rpm for 60 s and soft baking at 130 °C for 120 s, electron-beam exposure of alignment marks for both optical and electron-beam lithography at 10 nA with a shot pitch of 20 nm and 34 μC cm$^{-2}$ exposure dose. Post-exposure bake at 130 °C for 90 s, development in MF-24A (Microposit) for 90 s, rinsing in water and drying under N$_2$ stream. Reactive-ion etching (RIE) for 15 s at 60 mTorr chamber pressure, 60 W RF-power, 60 cm$^3$ STP min$^{-1}$ O$_2$ flow (PlasmaTherm Reactive Ion Etcher). RIE for 15 min at 40 mTorr chamber pressure, 50 W RF-power, 100 W ICP-power, 50 m$^3$ STP min$^{-1}$ Cl$_2$ flow (600 nm etch depth in silicon).

Thermal oxidation: Cleaning in 50 mL H$_2$O$_2$ and 100 mL H$_2$SO$_4$ at 130 °C for 10 min, rinsing in water and drying under a N$_2$ stream. Wet oxidation in water atmosphere for 45 min at 950 °C (200 nm thermal oxide).

Fabrication of nanochannels: spin coating HMDS adhesion promoter (MicroChem) at 3000 rpm for 30 s and soft baking on a hotplate at 115 °C for 120 s spin coating UV5 (MicroChem) at 2000 rpm for 60 s and soft baking at 130 °C for 120 s electron-beam exposure of nanochannels at 1 nA with a shot pitch of 10 nm and 34 μC cm$^{-2}$ exposure dose. Post-exposure bake at 130 °C for 90 s, development in MF-24A (Microposit) for 90 s, rinsing in water and drying under N$_2$ stream. RIE for 10 s at 40 mTorr chamber pressure, 40 W RF-power, 40 cm$^3$ STP min$^{-1}$ O$_2$ flow (PlasmaTherm Reactive Ion Etcher). RIE for 100 s at 8 mTorr chamber pressure, 50 W RF-power, 50 cm$^3$ STP min$^{-1}$ NF$_3$ flow (70 nm etch depth in thermal oxide). Cleaning in 50 mL H$_2$O$_2$ and 100 mL H$_2$SO$_4$ at 130 °C for 10 min, rinsing in water and drying under a N$_2$ stream.

Fabrication of microchannels: spin coating HMDS at 3000 rpm for 30 s and soft baking on a hotplate at 115 °C for 2 min. spin coating S1813 (Shipley) at 3000 rpm for 30 s and soft baking at 115 °C for 2 min. Expose microchannels for 10 s in a contact aligner at 6 mW cm$^{-2}$ intensity. Development in MF-319 (Microposit) for 60 s, rinsing in water and drying under N$_2$ stream. Buffered oxide-etch for 3 min to

remove thermal oxide, rinsing in water and drying under $N_2$ stream. Deep reactive-ion etching for 100 cycles of 7 s at 6 mTorr chamber pressure, 800 W RF-power, 8 W platen power, 130 cm$^3$ STP min$^{-1}$ SF$_6$ flow (Si-etch), and of 5 s at 6 mTorr chamber pressure, 800 W RF-power, 8 W platen power, 85 cm$^3$ STP min$^{-1}$ C$_4$F$_8$ flow (passivation) at a rate of 600 nm per cycle. Removal of resist in 50 mL $H_2O_2$ and 100 mL $H_2SO_4$ at 130 °C for 10 min, rinsing in water and drying under $N_2$ stream. The resulting channels have a depth of 60 μm measured using a Dektak 150 surface profiler.

Fabrication of inlets (from backside): Magnetron-sputtering of 200 nm Al (hard mask). Spin coating S1813 at 3000 rpm for 30 s and soft baking on a hotplate at 115 °C for 2 min. Expose inlets for 10 s in the contact aligner at 6 mW cm$^{-2}$ intensity. Development in MF-319 for 60 s, rinsing in water and drying under $N_2$ stream. Aluminum wet etch ($H_3PO_4$:$CH_3COOH$:$HNO_3$:$H_2O$ (4:4:1:1)) for 10 min to clear the hard mask at inlet positions. Deep reactive-ion etching for 300 cycles of 12 s at 5 mTorr chamber pressure, 600 W RF-power, 10 W platen power, 130 cm$^3$ STP min$^{-1}$ SF$_6$ flow (Si-etch), and of 7 s at 5 mTorr chamber pressure, 600 W RF-power, 10 W platen power, 85 cm$^3$ STP min$^{-1}$ C$_4$F$_8$ flow (passivation) at a rate of 2 μm per cycle. Removal of Al-hard mask in 50 mL $H_2O_2$ and 100 mL $H_2SO_4$ at 130 °C for 10 min, rinsing in water and drying under $N_2$ stream.

Fabrication of heater elements on the backside: Spin coating HMDS at 3000 rpm for 30 s and soft baking on a hotplate at 115 °C for 2 min. Spin coating LOR3A (MicroChemicals) at 3000 rpm for 30 s and soft baking at 180 °C for 5 min. Spin coating S1813 (Shipley) at 3000 rpm for 30 s and soft baking at 115 °C for 2 min. Expose heater elements with direct-laser lithography at 10 mW cm$^{-2}$ intensity. Development in MF-319 (Microposit) for 60 s, rinsing in water and drying under $N_2$ stream. Electron-beam evaporation of 10 nm Cr / 100 nm Pt. Lift-off in remover Rem1165 (MicroChemicals), rinsing in isopropanol, and drying under $N_2$ stream.

Fabrication of nanoparticles inside nanochannels: spin coating copolymer MMA(8.5)MMA (MicroChem Corporation, 10 wt.% diluted in anisole) at 6000 rpm for 60 s and soft baking on a hotplate at 180 °C for 5 min. spin coating ZEP520A: anisole (1:2) at 3000 rpm for 60 s and soft baking at 180 °C for 5 min. Electron-beam exposure at 1 nA with a shot pitch of 2 nm and 280 μC cm$^{-2}$ exposure dose. Development in $n$-amyl acetate for 60 s, rinsing in isopropanol and drying under $N_2$ stream. Development in methyl isobutyl ketone:isopropanol (1:1) for 60 s, rinsing in isopropanol and drying under a $N_2$ stream. Electron-beam evaporation of Au/SiO$_2$/Pd triple layer. Lift-off in acetone, rinsing in isopropanol, and drying under a $N_2$ stream.

Fusion bonding: (a) Cleaning of the substrate together with a lid (175 μm thick 4-inch-pyrex, UniversityWafers) in $H_2O$:$H_2O_2$:$NH_3OH$ (5:1:1) for 10 min at 80 °C. (b) Pre-bonding the lid to the substrate by bringing surfaces together and manually applying pressure. (c) Fusion bonding of the lid to the substrate for 5 h in $N_2$ atmosphere at 550 °C (°C min$^{-1}$ ramp rate).

Dicing of bonded wafers: Cutting nanofluidic chips from the bonded wafer using a resin-bonded diamond blade of 250 μm thickness (Dicing Blade Technology) at 35 krpm and 1 mm s$^{-1}$ feed rate.

## Data availability
The data that support the findings of this study are available from Zenodo[66] and from the corresponding author upon request. Source data are provided in this paper.

## Code availability
Code generated during this project, used to define and train the DAE algorithm, and to plot the results in this study is available from Zenodo[66] and from the corresponding author upon request.

Instructions for installation and proper usage is included in the documentation within the repository.

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

## Acknowledgements

This research has received funding from the European Research Council (ERC) under the European Union's Horizon Europe research and innovation program (101043480/NACAREI - C.L.) and under the European Union's Horizon 2020 research and innovation program (678941/SIN-CAT - C.L.), and from the Knut and Alice Wallenberg Foundation projects 2016.0210 (C.L.) and 2015.0055 (C.L.). Part of this work was carried out at the Chalmers MC2 cleanroom facility and at the Chalmers Materials Analysis Laboratory (CMAL). The authors also acknowledge the computer cluster Alvis, through which some of the computations were enabled by resources provided by the Swedish National Infrastructure for Computing (SNIC), partially funded by the Swedish Research Council through grant agreement no. 2022/22-386 and 2024/22-881.

## Author contributions

H.K.M., I.N., and C.L. conceived the project. J.F. and D.A. produced the nanofluidic chips. H.K.M., I.N., and G.A. performed the measurements. H.K.M. designed and implemented the DAE. H.K.M., G.A., and C.L. wrote the paper with input from all authors. C.L. provided the funding for the project.

## Funding

## Competing interests

The authors declare no competing interests.
