## [Transparent Peer Review file · Nature Communications]

Deep-learning-enabled online mass spectrometry of the reaction product of a single catalyst nanoparticle

Corresponding Author: Professor Christoph Langhammer

Version 0:

Reviewer comments:

Reviewer #1

(Remarks to the Author)

This is a very nice article, and although I do not understand all the deep learning aspects of the work; I admire the effort to perform the online mass spectrometry aspect, down to the level of a single nanoparticle surface, which is 3 order of magnitude smaller than normal work. The system has been tested for a model reaction and model system; namely CO oxidation over a Pd material. That of course makes that the work is a bit limited in its scope; and it would have been nice that the authors would have opted for a second reaction; for example a reduction process, which can still be done with e.g. the Pd system. Stated this, the authors have written a very impactful work and I recommend it with pleasure for publication in Nature Communications.

(Remarks on code availability)

Reviewer #2

(Remarks to the Author)

In this research, the authors have developed a method with the deep-learning by a constrained denoising auto-encoder to discern tiny signals from noise for the online mass spectrometry by connecting with the designed nanofluidic reactors. Taking the Pd model nanocatalysts in CO oxidation as an example, it was proved to be reduced by 3 orders of magnitude of surface area required for reactivity analysis compared to the current state of the art, and down to the level of the so-called single nanoparticle catalysis. With which, a stoichiometric highest reaction rate at the highest temperatures, with a distinct shift to lower concentration of CO at lower temperatures was confirmed, demonstrating the reliability of such method to acquire the microscopic mechanism in various catalytic reactions. In this regard, the authors have provided a promising method for online reaction analysis even in a single nanoparticle (thousands of nm² in surface area) as a detection limit with mass spectrometry techniques. It is interesting, which would inspire relevant studies towards mechanism understanding to avoid the mean approximation of nanoparticle ensembles. Therefore, I would like to recommend its publication. Some suggestions are provided below for the authors' consideration in the revision process.

1. The authors have given the dimensions of their chip of reactors in Figure 1B, while the height of this reactors is only 200 nm. In such a nanofluidic reactor, will the mobility or transfer of gases show some different characters as compared with that in the conventional reactors? This might be an interesting and important topic in this nanofluidic reactor.
2. From the counting results for n= 1000, 10, and 1, a statistical rate of Pd nanoparticles in CO oxidation can be achieved for the catalysts. Moreover, is it possible to quantify the CO₂ concentration by the count number, by which an exact TOF in the CO oxidation can be obtained in consideration of the surface area.
3. In Figure 4, the counts for Pd₁₀ after DAE has a decrease as compared with the original data. Does it mean the underestimation of CO₂ production after DAE?
4. As seen in Figure 5, it shows that the quality of data can be great improved after DAE-based deep learning to discern signals from noise, however, the counts for Pd₁ nanoparticle in 5f was only several numbers, will it lead to an error in these counts which would have influence on the results of activity analysis?
5. The CO oxidation over Pd nanoparticle might be not a good case in this single particle catalysis due to its inferior activity. The catalytic system with a high reactivity would be a more promising case in the nanofluidic reactor combined with the online mass spectrometry by DAE analysis.
6. The references in the article, for example, ref 39, seems to be incorrect, and the Figure citing in Page 10, (Figure 1E and

Figure 1F) would be Figure 1F and Figure 1G, respectively.

(Remarks on code availability)

Reviewer #3

(Remarks to the Author)

According to its title, this article promises a general solution for improving online mass spectrometry. However, the content of the article describes a specialized model study on a single well-known reaction of CO oxidation and applied to a single signal. There is no evidence that the described methodology has been tested on different catalytic reactions. Nor is it shown that the methodology described is applicable to a general method for distinguishing low intensity signals from noise.

The authors' conclusion "These results constitute a new paradigm for online reaction analysis in single particle catalysis and advocate deep learning to improve resolution in mass spectrometry in general" is a drastically overstated message that is not related to the actual results described.

The methodology of online mass spectrometry, as claimed in the study, would imply that the authors should develop software and make it available to other researchers to improve signal-over-noise detection in general. After reading the article, I did not find any evidence of general practical use.

Overall, the study is too specialized for a high impact journal. The article may be considered for publication in Communications Chemistry if the model, program, and dataset are made publicly available upon publication of the article.

(Remarks on code availability)

Version 1:

Reviewer comments:

Reviewer #1

(Remarks to the Author)

I have read the revised article as well as the rebuttal letter in detail. I am somewhat happy that the different referees come to the same conclusion and that based on this the authors have opted for another catalytic reaction based on their catalyst system and therefore also broadened their study beyond the CO oxidation reaction. I also majorly agree now with the statements and finding reported and also believe the authors have done a really good job to address the major concerns by me and the other two referee reports. Having stated this, I find still some sentences rather bold and overstated and would somewhat tone down some of the aspects on the impact and novelty of their work. It is for me not needed to overstate this as the study is ultimately published in an impactful journal Nature Communications and hence there is no need to do this. Hence, I would suggest the authors to carefully go through their manuscript again and see where possible to make the article more "grounded". I also wish to state that the use of microreactor technology in the field of heterogeneous catalysis and analyzing it at the level of a single particle is not entirely new and that the authors may see how their work relates to this single particle analysis approaches, which were not focusing on mass spectrometry but on fluorescence detection. Having stated this, I am fully supported this work to be published in Nature Communications.

(Remarks on code availability)

Not assessed.

Reviewer #2

(Remarks to the Author)

The authors have carefully answered the questions requested by reviewers, with additional experimental evidences to support their conclusion. I have no more suggestions, and recommend its publication in the Journal.

(Remarks on code availability)

Reviewer #3

(Remarks to the Author)

1) Catalyst preparation/characterization (SI, page 5; and Fig. 1 in the article) is difficult to understand. The authors mark low contrast gray areas (appearing very similar to each other) as Pd, SiO₂ and Au (Figure S2). This image does not look like a metal particle. Pd and Au particles in electron microscopy should appear as high contrast objects. I do not see any direct evidence for metal nanoparticles in this image. There is also no way to distinguish between Pd and Au with these analytical data.

Summary of points on the catalyst:

- a) catalyst preparation procedure should be described in sufficient details for the reproduction;
- b) better quality electron microscopy images should be provided;
- c) proper catalyst characterization should be provided;
- d) "n = 1000 Pd nanoparticle sample": If a researcher wants to reproduce the synthesis of "n = 1000 Pd nanoparticle sample" what the procedure is?
- e) experimental evidence and analytics for the presence of 1000 Pd nanoparticles should be provided.
- f) For each sub-image or insert on Fig. 1 in the article, the corresponding full-scale image should be provided in the SI, with the description allowing to reproduce the sample preparation and the measurements.

2) The results described in this study are highly dependent on the accuracy of the quadrupole mass spectrometer (QMS) measurements. What is the actual accuracy of the measurements used? An appropriate description should be provided.

3) In fact, the results reported in this article concern ML-based attenuation of a signal, or the distinguishing of low-intensity signals from noise. The role of the catalyst is to produce a low intensity signal. However, there is no evidence that the procedure described by the authors measures the actual signals. There is no independent evidence. Finding artifact/phantom signals is a well-known case when searching for low-intensity signals computationally.

Independent measurements should be easy to perform – one can analyze a low concentration sample of known composition. Low concentration samples (in the same concentration range as in the catalytic reaction being studied) should be prepared by mixing/diluting appropriate gases. The calibration curve should be reported to demonstrate that the ML-enhanced method adequately measures the signals.

4) The article contains statements like "see Methods for explanation of the BA-procedure", but it is difficult to locate the corresponding places in the description. The sections should be numbered and cited by the section number in the text (both – sub/sections in the articles and SI), while citation of unclear locations should be avoided.

5) Statements about "a new paradigm" are drastically overstated and do not relate to the actual results described.

6) The authors state "to significantly improve the resolution", which is unclear. Resolution is the ability to distinguish signals appearing close to each other. While registering low intensity signal refers to sensitivity. Sensitivity and resolution in MS are different terms.

7) "We have demonstrated how the combination of nanofluidic reactors and DAE-based deep learning enables the reduction of catalyst surface area required for online QMS analysis of reaction product in the gas phase by ≈ 3 orders from the current SotA, down to the level of single nanoparticles." - in fact, this statement means that the sensitivity of the analysis was improved and the measurements for a low concentration gas should be possible.

While "reduction of catalyst surface" and "down to the level of single nanoparticles" have no direct connection with the actual ML development made. These are possible areas of application of the sensitivity enhancement, rather than direct contribution to catalysis. It should be clearly stated that the article in its present form makes no contribution to the catalysis/nanoparticles topics. The results described concern sensitivity enhancements in gas analysis that (possibly) will make a contribution in catalysis/nanoparticles topics in the future. Misleading statements should be avoided.

(Remarks on code availability)

Version 2:

Reviewer comments:

Reviewer #1

(Remarks to the Author)

I have read the rebuttal letter and also assessed the revisions made in the revised article in response to the different referees. Based on all this, I am happy with the revised version and hence recommend the work for publication in Nature Communications.

(Remarks on code availability)

Reviewer #3

(Remarks to the Author)

The authors have addressed most of the comments by additional experiments and text improvements. Some of the suggested changes were not fully implemented, but the authors have provided satisfactory response to the comments. The article can be considered for publication.

(Remarks on code availability)

Point-to-Point Response NCOMMS-24-14980-T

Reviewer #1

This is a very nice article, and although I do not understand all the deep learning aspects of the work; I admire the effort to perform the online mass spectrometry aspect, down to the level of a single nanoparticle surface, which is 3 order of magnitude smaller than normal work. The system has been tested for a model reaction and model system; namely CO oxidation over a Pd material. That of course makes that the work is a bit limited in its scope; and it would have been nice that the authors would have opted for a second reaction; for example a reduction process, which can still be done with e.g. the Pd system. Stated this, the authors have written a very impactful work and I recommend it with pleasure for publication in Nature Communications.

Our reply: We thank the reviewer for the very positive feedback and the suggestion to extend our study to another reaction to widen its scope, with which we completely agree. We have therefore incorporated this suggestion into our manuscript in the form of a new section added in the end, where we investigate the ethylene hydrogenation reaction on 1000 Pd nanoparticles using an inferior QMS in terms of sensitivity, compared to the CO oxidation experiments. Our reasoning behind this selection of reaction, catalyst bed and QMS was the following: (i) use a hydrogenation reaction to contrast the oxidation reaction already demonstrated; (ii) use a reaction with higher activity than CO oxidation, as suggested by Reviewer #2; (iii) use an inferior QMS with lower intrinsic sensitivity to demonstrate the idea of using the DAE data analysis as a means to boost the performance of a QMS one may have at hand in the lab but that is not really “fit for the job” even in a less extreme scenario than catalysis on a single nanoparticle – as exemplified on the ethylene hydrogenation reaction on a 1000 Pd nanoparticle catalyst bed.

Reviewer #2 (Remarks to the Author):

In this research, the authors have developed a method with the deep-learning by a constrained denoising auto-encoder to discern tiny signals from noise for the online mass spectrometry by connecting with the designed nanofluidic reactors. Taking the Pd model nanocatalysts in CO oxidation as an example, it was proved to be reduced by 3 orders of magnitude of surface area required for reactivity analysis compared to the current state of the art, and down to the level of the so-called single nanoparticle catalysis. With which, a stoichiometric highest reaction rate at the highest temperatures, with a distinct shift to lower concentration of CO at lower temperatures was confirmed, demonstrating the reliability of such method to acquire the microscopic mechanism in various catalytic reactions. In this regard, the authors have provided a promising method for online reaction analysis even in a single nanoparticle (thousands of nm² in surface area) as a detection limit with mass spectrometry techniques. It is interesting, which would inspire relevant studies towards mechanism understanding to avoid the mean approximation of nanoparticle ensembles. Therefore, I would like to recommend its publication. Some suggestions are provided below for the authors' consideration in the revision process.

Our reply: We would like to sincerely thank the reviewer for her/his overall positive assessment of our work and for the constructive comments that we address in detail below.

1. The authors have given the dimensions of their chip of reactors in Figure 1B, while the height of this reactors is only 200 nm. In such a nanofluidic reactor, will the mobility or transfer of gases show some different characters as compared with that in the conventional reactors? This might be an interesting and important topic in this nanofluidic reactor.

Our reply: The reviewer is correct that the reactor height of 200 nm implies that the gas flow is rarefied in the nanochannel. More specifically, the flow is in the slip flow and transitional flow regimes, with the Knudsen number increasing in the mean flow direction throughout the channel, as we also have discussed in our previous work using a unified channel flow model (Science Advances **2020**, 6 (25), eaba7678; Nature Communications **2020**, 11 (1), 4832). As a consequence, momentum, heat and mass transfer boundary layers surrounding reacting particles are thinner than in the continuum regime, and the mass diffusivities approach the Knudsen diffusivity. Importantly, these effects are all present in technically widely relevant porous catalyst systems, indicating that the nanofluidic reactor offers a way to investigate heterogenous chemistry in what we earlier have coined "model pores" under controlled conditions. To make this clear we have added the following text to the revised manuscript:

*"Furthermore, we note that the gas flow in such a system is in the slip flow and transitional flow regimes, with the Knudsen number increasing in the mean flow direction throughout the nanofluidic reactor (Nature Communications **2020**, 11 (1), 4832). Consequently, momentum, heat and mass transfer boundary layers surrounding reacting particles are thinner than in the continuum regime characteristic for conventional reactors, and the mass diffusivities approach the Knudsen diffusivity. In fact, nanofluidic reactors therefore constitute effective "model pores" since the aforementioned effects all are present in technically widely used porous catalyst systems (Science Advances **2020**, 6 (25), eaba7678)."*

2. From the counting results for n= 1000, 10, and 1, a statistical rate of Pd nanoparticles in CO oxidation can be achieved for the catalysts. Moreover, is it possible to quantify the CO₂ concentration by the count number, by which an exact TOF in the CO oxidation can be obtained in consideration of the surface area.

Our reply: In principle, the reviewer is correct that one should be able to extract a ToF from the QMS measurements. However, to do so one has to calibrate the QMS signal for the CO₂ reaction product. We have, unfortunately, not done such a calibration and since the CO oxidation experiments have been executed the used QMS has been serviced by the supplier (e.g. the filament of the ion source has been exchanged) during the period we had mounted the inferior QMS used for the ethylene hydrogenation measurements since that was a perfect opportunity to do so without impacting ongoing work. Therefore, the QMS is now in a different state and we are unable to do such a retroactive calibration and thus we can, unfortunately, not provide any ToFs.

3. In Figure 4, the counts for Pd10 after DAE has a decrease as compared with the original data. Does it mean the underestimation of CO₂ production after DAE?

Our reply: The decrease in counts after DAE is not an underestimation of CO₂ production. Instead, it reflects the DAE's ability to remove noise and improve the signal-to-noise ratio. We have clarified this point in the discussion of Figure 4 by adding the following sentence:

“... Thus, the measured raw value of 90 counts likely overestimates the true CO₂ production rate due to residual noise or background signal, highlighting the DAE's ability to accurately correct for that.”

4. As seen in Figure 5, it shows that the quality of data can be great improved after DAE-based deep learning to discern signals from noise, however, the counts for Pd1 nanoparticle in 5f was only several numbers, will it lead to an error in these counts which would have influence on the results of activity analysis?

Our reply: We agree with the reviewer that the low counts for Pd1 nanoparticles in principle could lead to an error. However, we believe that the DAE's ability to discern signals from noise mitigates this issue to a large extent. We have added a discussion of this point in the revised manuscript as follows:

“As a last point, we note that the low absolute counts observed for single nanoparticles (c.f. Pd1 in Figure 5F) raise the question of potential error in the activity analysis in this regime of very few counts. While it is true that low count numbers can, in principle, lead to larger relative uncertainties, the DAE's denoising capabilities mitigates this issue. Specifically, the DAE is trained to distinguish genuine signals from noise, even when the signal is very weak. Therefore, while the absolute number of counts might be low, the proportion of those counts representing actual CO₂ production is very high after DAE processing compared to the raw data and thus the counts predicted by the DAE are indeed significant. Furthermore, the consistency of the DAE-derived trends across multiple independent measurements (c.f. Figure 5F) corroborates the reliability of the activity trends obtained by the DAE, even for single nanoparticles.”

5. The CO oxidation over Pd nanoparticle might be not a good case in this single particle catalysis due to its inferior activity. The catalytic system with a high reactivity would be a more promising case in the nanofluidic reactor combined with the online mass spectrometry by DAE analysis.

Our reply: We thank the reviewer for this comment and following his/her advice, we selected the hydrogenation of ethylene over a Pd catalyst as the reaction of choice for a second catalysis example that we have added as a new section at the end of the revised manuscript (to also fulfill the corresponding requests by Reviewer #1 and #3). This specific selection of reaction was made on the basis that: (i) a hydrogenation reaction nicely contrast the already demonstrated oxidation reaction to showcase the wide applicability of our nanofluidic reactor concept in terms of catalytic reactions; (ii) the hydrogenation of ethylene on Pd indeed is significantly faster than the oxidation of CO.

6. *The references in the article, for example, ref 39, seems to be incorrect, and the Figure citing in Page 10, (Figure 1E and Figure 1F) would be Figure 1F and Figure 1G, respectively.*

Our reply: We apologize for the errors in the references and figure citations. We have corrected these mistakes in the revised manuscript.

Reviewer #3

1. According to its title, this article promises a general solution for improving online mass spectrometry. However, the content of the article describes a specialized model study on a single well-known reaction of CO oxidation and applied to a single signal. There is no evidence that the described methodology has been tested on different catalytic reactions. Nor is it shown that the methodology described is applicable to a general method for distinguishing low intensity signals from noise.

Our reply: As already outlined in our replies to Reviewers #1 & #2, we have executed a second study on the hydrogenation of ethylene on a Pd catalyst to indeed demonstrate that our concept – both the experimental nanoreactors and the DAE data treatment – are applicable to different catalytic reactions and thus in this sense generic. This new study is incorporated into our revised manuscript in the form of a new section added in the end. Beyond introducing a second catalytic reaction, this section also explicitly highlights the general applicability of the DAE to extract small signals from noise since the limiting factor in these experiments was not the amount of catalytic surface area but rather the intrinsic sensitivity of the QMS, which we had exchanged to a less sophisticated model than the one used for the CO oxidation experiments to emulate the very common scenario of an available experimental hardware not delivering the required performance for a challenging experiment – and to then demonstrate how the use of the denoising autoencoder model constitutes a means to overcome this generic hardware limitation.

As a second aspect, we also again highlight that the underlying principles of our method, particularly the denoising autoencoder, are inherently generalizable beyond the specific examples presented. The strength of our method lies in its ability to extract weak signals from noisy data, regardless of the specific reaction being studied (or for that matter, which method that generated the data). This initial demonstration, now including two distinct catalytic reactions, thus indeed provides a robust foundation for future studies in the realm of weak catalytic signals and the use of machine learning models to extract these weak signals from noise.

2. The authors' conclusion "These results constitute a new paradigm for online reaction analysis in single particle catalysis and advocate deep learning to improve resolution in mass spectrometry in general" is a drastically overstated message that is not related to the actual results described.

Our reply: We acknowledge the reviewer's concern regarding the scope of our concluding statement. However, the conclusion, as written, reflects our perspective on the broader implications of this work, which we now are convinced have been significantly further strengthened with the inclusion of the results from a second catalytic reaction in the revised manuscript. Hence, we find it accurate and not overstated, in particular also because the fundamental principles underlying our denoising autoencoder are not inherently limited to the specific reactions studied in this work as examples, and hold real promise for broader applicability in mass spectrometry in catalysis and beyond.

3. The methodology of online mass spectrometry, as claimed in the study, would imply that the authors should develop software and make it available to other researchers to improve signal-

over-noise detection in general. After reading the article, I did not find any evidence of general practical use.

Our reply: We can acknowledge the reviewer's point regarding the development of a generally applicable software package. However, we argue that a full software suite is beyond the scope of this initial demonstration, and we have prioritized reproducibility. Furthermore, we indeed provide the code necessary to implement our deep learning-based denoising approach, including scripts for setup, training, and analysis of the denoising autoencoder. This code provides researchers with the essential tools to reproduce and adapt our methodology. We indeed believe this level of transparency is crucial for fostering further development and broader application of this technique. Future work will focus on packaging these tools into a more user-friendly software environment.

4. Overall, the study is too specialized for a high impact journal. The article may be considered for publication in Communications Chemistry if the model, program, and dataset are made publicly available upon publication of the article.

Our reply: We hope that our responses to all Reviewer comments, including reviewer #1 & #2) have been able to convince Reviewer #3 of the broad relevance and impact of our work. We believe that the revised manuscript constitutes a significant further improvement in terms of scope and relevance and that will be of interest to the readers of Nature Communications.

Point-to-point response

Reviewer #1 (Remarks to the Author):

1. I have read the revised article as well as the rebuttal letter in detail. I am somewhat happy that the different referees come to the same conclusion and that based on this the authors have opted for another catalytic reaction based on their catalyst system and therefore also broadened their study beyond the CO oxidation reaction.

Our reply: We are happy the Reviewer acknowledges that we have been able to broaden our study as requested.

2. I also majorly agree now with the statements and finding reported and also believe the authors have done a really good job to address the major concerns by me and the other two referee reports.

Our reply: We thank the Reviewer for acknowledging our efforts to address all major concerns.

3. Having stated this, I find still some sentences rather bold and overstated and would somewhat tone down some of the aspects on the impact and novelty of their work. It is for me not needed to overstate this as the study is ultimately published in an impactful journal Nature Communications and hence there is no need to do this. Hence, I would suggest the authors to carefully go through their manuscript again and see where possible to make the article more "grounded".

Our reply: We apologize for using wordings that are perceived as overstating and have now made an effort to amend this wherever we ourselves identified sentences that could be perceived as overstated.

4. I also wish to state that the use of microreactor technology in the field of heterogeneous catalysis and analyzing it at the level of a single particle is not entirely new and that the authors may see how their work relates to this single particle analysis approaches, which were not focusing on mass spectrometry but on fluorescence detection.

Our reply: We agree with the Reviewer that there exists a body literature on microreactors in catalysis. However, we would like to emphasize that we refer to our systems as nanoreactors, since they height/thickness of the catalyst bed is only 200 nm. That said, we also explicitly refer to and cite key microreactor literature in the introduction, where it reads:

"To address this limitation, miniaturization of the reactor to the level of microreactors [36–38] and even nanoreactors [40] to reduce the catalyst surface area necessary to obtain measurable (QMS) signals has been implemented. Very high sensitivity can be obtained in such systems by dramatically reducing reactor volume and thereby directing the entire gas flow from the catalyst bed to a QMS to maximize the fraction of reaction product available for analysis. In this way, as we recently have demonstrated using a single nanofluidic channel as the catalyst bed, the minimal required active catalyst surface area for online QMS measurements was reduced by ca. 1.5 orders compared to the lower limit proposed for microreactors [36], i.e., down to ca. 10 μm² active surface area [41]."

Accordingly, we do not see any need to further elaborate on the connection of our current work to the field of microreactors.

When it comes to single particle experiments using fluorescence, we agree that these studies indeed have pioneered the field of single particle catalysis, however, arguably together with other methods such as tip-enhanced Raman spectroscopy or SHINERS. We indeed refer to all these pioneering methods in the introduction in the following sentences:

"To this end, in the most extreme implementation, the aim of so-called single particle catalysis is to study catalytic reactions on individual nanoparticles to overcome ensemble averaging effects [30]. The first strategy to enable such experiments employs detection of photon or electron signals that report on either the product molecules formed, on reactant molecules consumed, on the catalyst particle itself, or on temperature changes generated by the reaction [30–35]. While both elegant and effective for the specific reactions chosen, these approaches lack the wide and generic applicability of mass spectrometry. Furthermore, they often cannot deliver information about both amounts and molecular weight of the species involved in the reaction."

As a second aspect, we also want to highlight here that all these single particle catalysis studies, fluorescence in particular, are all executed in the liquid phase rather than in the gas phase, as we do in our work here. Hence, we argue that in more depth than we already do introducing these efforts is not critically relevant to put our work into context. Nevertheless, we have added the following sentence and references to the introduction of the revised manuscript to explicitly highlight fluorescence-based single particle catalysis due to its pioneering role:

“To this end, in the most extreme implementation, the aim of so-called single particle catalysis is to study catalytic reactions on individual nanoparticles to overcome ensemble averaging effects [30]. The first strategy to enable such experiments employs detection of photon or electron signals that report on either the product molecules formed, on reactant molecules consumed, on the catalyst particle itself, or on temperature changes generated by the reaction [30–35]. In this context, single molecule fluorescence microscopy stands out due to its ability to resolve individual reaction events on single catalyst nanoparticles with very high spatial resolution ([29] Zhou, X. et al. Quantitative super-resolution imaging uncovers reactivity patterns on single nanocatalysts. Nature Nanotechnology 7, 237–241 (2012). [30] Xu, W., Kong, J. S., Yeh, Y.-T. E. & Chen, P. Single-molecule nanocatalysis reveals heterogeneous reaction pathways and catalytic dynamics. Nature Materials 7, 992–996 (2008). [31] Chen, P., Xu, W., Zhou, X., Panda, D. & Kalininskiy, A. Single-nanoparticle catalysis at single-turnover resolution. Chemical Physics Letters 470, 151–157 (2009).). While both elegant and effective for the specific reactions chosen, however, these approaches are compatible only with catalysis in the liquid phase and lack the wide and generic applicability of mass spectrometry. Furthermore, they often cannot deliver information about both amounts and molecular weight of the species involved in the reaction.”

Having stated this, I am fully supported this work to be published in Nature Communications.

Our reply: We thank the Reviewer for supporting our work to be published in Nature Communications.

Reviewer #2 (Remarks to the Author):

The authors have carefully answered the questions requested by reviewers, with additional experimental evidences to support their conclusion. I have no more suggestions, and recommend its publication in the Journal.

Our reply: We thank the Reviewer for recommending our work for publication.

Reviewer #3 (Remarks to the Author):

1) Catalyst preparation/characterization (SI, page 5; and Fig. 1 in the article) is difficult to understand. The authors mark low contrast gray areas (appearing very similar to each other) as Pd, SiO₂ and Au (Figure S2). This image does not look like a metal particle. Pd and Au particles in electron microscopy should appear as high contrast objects. I do not see any direct evidence for metal nanoparticles in this image. There is also no way to distinguish between Pd and Au with these analytical data.

Our reply: We can agree with the Reviewer that the resolution of or SEM images is not perfect but at the same time want to emphasize that this simply is the consequence of the fact that the catalyst particles are nanofabricated onto a thermally oxidized silicon wafer, which means they sit on a layer of insulating SiO₂. This in turn leads to charging effects during SEM imaging that reduces resolution. As a consequence, it is almost impossible to obtain better SEM images than the ones we have in our work. When it comes to distinguishing between the different materials, we argue that it indeed is possible to distinguish the two different metal particles and the (“invisible”) SiO₂ due to low contrast in between in the side-view image in Figure 1C. Finally, as discussed in detail below in response to further comments on this topic, we emphasize that we do not synthesize our nanoparticles but put them in place with nm spatial accuracy using nanolithography techniques and thin film deposition which ensures (i) that particles indeed are where we want them and (ii) are comprised of exactly the materials and layers we predefine, with Å resolution in terms of thickness of the deposited layers. Hence, we are very confident that Au indeed is Au, SiO₂ indeed SiO₂ and Pd indeed Pd since they have been deposited layer-by-layer through the lithography mask using physical vapor deposition.

Summary of points on the catalyst:

a) catalyst preparation procedure should be described in sufficient details for the reproduction;

Our reply: We describe in detail all the steps of the catalyst nanoparticle fabrication in the Methods sub-section entitled “Fabrication of nanoparticles inside nanochannels”. Furthermore, we have added the following clarifying text to the revised manuscript:

“Using electron-beam lithography (EBL) nanofabrication (1), they were decorated...., as SEM image analysis reveals, physical vapor deposited (PVD) onto an 8 nm thick SiO₂ support layer that separates them from chemically inert plasmonic Au nanoparticles previously PVD-grown through the same EBL mask with....”

b) better quality electron microscopy images should be provided;

Our reply: As already note above, this is not possible due to the nature of the substrate used due to charging effects. This is especially pronounced for tilted side-view images.

c) proper catalyst characterization should be provided;

Our reply: We are not sure what specific characterization the Reviewer is expecting here but as argued above, it is very difficult to execute other characterization than that one we have provided as a consequence of the nature of the sample. We also reiterate that we are very confident in the EBL nanofabrication method that we use to deliver the exact nanostructures that we want with very high accuracy (Chen, Y., Nanofabrication by electron beam lithography and its applications: A review. *Microelectronic Engineering* 2015, 135, 57-72). In essence, we use the EBL tool to explicitly define the position and size of each catalyst particle with nanometer spatial resolution when preparing the mask. Hence, we can exactly define the number of catalyst particles in this way. Subsequently, once the mask has been developed, we use physical vapor deposition through the mask to grow the particles at the predefined positions. The thickness of the evaporated material is controlled with Ångström resolution. Hence, using this process, the obtained nanoparticles are very well defined in terms of their exact position, size and thickness. Finally, we also note that we obtain the Au/SiO₂/Pd structures by subsequently evaporating Au, SiO₂ and finally Pd layer-by-layer through the mask, thereby accurately controlling the material composition of each particle.

d) “n = 1000 Pd nanoparticle sample”: If a researcher wants to reproduce the synthesis of “n = 1000 Pd nanoparticle sample” what the procedure is?

Our reply: The procedure is described in detail in the corresponding Methods section entitled “Fabrication of nanoparticles inside nanochannels”. As stated above, we do not use any synthesis methods (such as colloidal synthesis in solution) to make our particles. Instead, we rely on EBL which is a very well-established methodology that explicitly “writes” each particle. Hence, reproducing N=1000 (or any other desired value of N) is indeed straightforward.

e) experimental evidence and analytics for the presence of 1000 Pd nanoparticles should be provided.

Our reply: We have provided experimental evidence for the presence of the particles on each sample, not only N=1000, in Figure 1 D-F by showing optical dark-field scattering microscopy images taken through the sealed (with glass lid for optical transparency) chip. In these images each bright spot corresponds to a single nanoparticle in the array. As we have shown in our earlier work, each particle can be tracked individually by its optical signature, if desired (*Nature Communications* 2020, 11 (1), 4832). In fact, as stated in the manuscript, we have added the Au plasmonic nanoantennae to the system to enable the dark-field scattering microscopy-based verification of the presence of each individual particle in the nanoreactors. This, in combination with the high reliability of the used EBL nanofabrication method used to make the particle array, we argue, is evidence enough for the presence of the particles (together with the fact that the experimentally measured catalyst activity indeed nicely scales with number of particles, N). Finally, we also want to highlight that single particle “counting” with other methods is very difficult/impossible when the fluidic chip is sealed as it then, e.g., is no longer transparent for electrons which makes electron microscopy impossible.

f) For each sub-image or insert on Fig. 1 in the article, the corresponding full-scale image should be provided in the SI, with the description allowing to reproduce the sample preparation and the measurements.

Our reply: First, we highlight that most images in Figure 1 are dark-field light scattering microscopy images and not SEM. In our opinion, providing more zoomed-out images would not really make sense as one then simply would not be able to resolve anything in these images. When it comes to the shown SEM images, we have added a further zoomed-out SEM image to the SI as a new figure (also pasted below for convenience). It shows an array of N= 1000 particles confirming indeed their presence prior to the bonding of the glass lid needed for hermetically sealing of the nanofluidic chip. For the tilted image of a single particle in panel C that shows the

layered Au/SiO₂/Pd structure, we note that the zoomed-out version already is included in the SI as Supplementary Figure 3. For the latter image, we again note that the blurriness is the consequence of the insulating substrate that makes taking tilted high resolution SEM images very challenging and it is impossible for us to obtain better images, in particular at high resolution and from a tilt angle. When it becomes to the description of how the samples were prepared and how the experiments were conducted, we refer to the Methods section where these details indeed are provided.

Figure caption SI Figure 3: zoomed-out SEM of a chip showing the full array of N=1000 Pd nanoparticles. Top-view SEM micrograph of the nanofluidic channel region (bounded by the two bright horizontal lines) displaying the complete array of 1000 Pd nanoparticles (nominal diameter 60 nm, center-to-center pitch 270 nm) grown through the lithographic mask. The uniform particle placement across the entire channel area confirms the successful templating of Pd via physical-vapor deposition. Scale bar: 2 microns.

The Figure presents a zoomed-out top-view of the N=1000 Pd chip, where the two bright horizontal bands mark the channel walls and the intervening field contains the full grid of N=1000 individual Pd nanoparticles. Each bright dot in the array corresponds to a single Pd catalyst particle, demonstrating uniform size, spacing, and coverage across the channel. This large-area SEM overview verifies the precise control over nanoparticle templating.

2) The results described in this study are highly dependent on the accuracy of the quadrupole mass spectrometer (QMS) measurements. What is the actual accuracy of the measurements used? An appropriate description should be provided.

Our reply: We appreciate the reviewer's point and have added a sentence in the Methods section about protocol. While we did not perform absolute calibration for quantitative analysis, the QMS was operated under stable and reproducible conditions with fixed flow rates, temperature, and acquisition settings. For our study, which focuses on resolving weak signals using the DAE, reproducibility and relative signal fidelity are more critical than absolute accuracy. Background subtraction was applied systematically, and all experiments were performed with identical gas handling and detection configurations to ensure comparability. The added Methods section clarifies these points. We have added the following to the revised Methods:

"No absolute calibration was performed; instead, all signals were analyzed comparatively using consistent instrument settings, flow conditions, and background subtraction protocols for all executed experiments."

3) In fact, the results reported in this article concern ML-based attenuation of a signal, or the distinguishing of low-intensity signals from noise. The role of the catalyst is to produce a low intensity signal. However, there is no evidence that the procedure described by the authors measures the actual signals. There is no independent evidence. Finding artifact/phantom signals is a well-known case when searching for low-intensity signals computationally. Independent measurements should be easy to perform – one can analyze a low concentration sample of known composition. Low concentration samples (in the same concentration range as in the catalytic reaction being studied) should be prepared by mixing/diluting appropriate gases. The calibration curve should be reported to demonstrate that the ML-enhanced method adequately measures the signals.

Our reply: We thank the reviewer for this important and constructive comment. In response, we have conducted additional control experiments to confirm that the signals reconstructed using the DAE method correspond to real, low-intensity CO₂ signals rather than computational artifacts. Specifically, we performed a set of experiments using a nanofluidic chip identical in geometry to that used in the catalytic measurements but deliberately fabricated without any Pd nanoparticles or catalytically active material. Pulses of CO₂ diluted in Ar were introduced under the same operating conditions (450 °C, 2 bar, 20 mL/min flow), and the resulting QMS signal was analyzed. The DAE-denoised output successfully reconstructed the timing and profile of the CO₂ pulses, even when the raw signal faded into the noise, providing independent confirmation of the method's ability to recover genuine low-intensity signals. This is described in the main text in the following newly added section:

“To ensure that the CO₂ signal observed in the catalytic experiments originates exclusively from surface reactions on Pd nanoparticles, we conducted a control experiment using an identical nanofluidic chip lacking any catalytically active material. Pulses of CO₂ diluted in Ar were flushed through the chip under identical conditions (450 °C, 2 bar, 20 ml/min total flow), and the resulting QMS signal was analyzed. As detailed in Figure 17A–B in the SI, no evidence of spurious CO₂ formation or retention effects was observed. Furthermore, the DAE successfully retrieved the shape and timing of the low-intensity CO₂ pulses even when the raw signal approached the noise floor, confirming the robustness of the signal processing method and excluding non-catalytic sources of CO₂ response.”

A more detailed account of the experimental setup and data from these additional experiments are provided in the revised Supplementary Information, in the new Section: “Control Experiment: Detection of Pulsed CO₂ in the Absence of Pd Nanoparticles” and Supplementary Figure 17A–B, which is pasted below for convenience.

Supplementary Figure 17: Validation of CO₂ Signal Origin and DAE Performance in a Non-Catalytic Nanofluidic Chip. (A) Flow profile showing the introduction of discrete CO₂ pulses (diluted in Ar) through a nanofluidic chip with no Pd or catalytically active material. The pulse sequence spans a range of CO₂ concentrations, progressively decreasing to test the detection limit. (B) QMS signal at $m/z = 44$ after background subtraction (grey), overlaid with the output of the DAE-denoising algorithm (purple). Despite the low intensity of the raw signal at lower concentrations, the denoised trace successfully reconstructs the timing and shape of the CO₂ pulses. This demonstrates the robustness of the DAE approach for signal recovery in high-noise regimes.

4) The article contains statements like “see Methods for explanation of the BA-procedure”, but it is difficult to locate the corresponding places in the description. The sections should be numbered and cited by the section number in the text (both – sub/sections in the articles and SI), while citation of unclear locations should be avoided.

Our reply: We have tried to improve this by numbering sections in Methods.

5) Statements about “a new paradigm” are drastically overstated and do not relate to the actual results described.

Our reply: We have reworded accordingly.

6) The authors state “to significantly improve the resolution”, which is unclear. Resolution is the ability to distinguish signals appearing close to each other. While registering low intensity signal refers to sensitivity. Sensitivity and resolution in MS are different terms.

Our reply: We thank the Reviewer for pointing this out and have reworded from “resolution” to “sensitivity”.

7) “We have demonstrated how the combination of nanofluidic reactors and DAE-based deep learning enables the reduction of catalyst surface area required for online QMS analysis of reaction product in the gas phase by \approx 3 orders from the current SotA, down to the level of single nanoparticles.” - in fact, this statement means that the sensitivity of the analysis was improved and the measurements for a low concentration gas should be possible. While “reduction of catalyst surface” and “down to the level of single nanoparticles” have no direct connection with the actual ML development made. These are possible areas of application of the sensitivity enhancement, rather than direct contribution to catalysis. It should be clearly stated that the article in its present form makes no contribution to the catalysis/nanoparticles topics. The results described concern sensitivity enhancements in gas analysis that (possibly) will make a contribution in catalysis/nanoparticles topics in the future. Misleading statements should be avoided.

Our reply: While we agree with the reviewer that our work describes a sensitivity enhancement in gas analysis we do not agree that “it makes no contribution to the catalysis/nanoparticles topics”. In fact, a second key aspect of our work that we highlight clearly (Introduction: “To overcome this gap we employ here a new type of nanofluidic reactor and combine it with a constrained convolutional denoising auto-encoder. This harnesses the reaction product focusing capability of nanofluidic reactors with the versatility, high-resolution and sensitivity of the QMS, with the power of deep learning to detect and analyze very weak signals hidden in noise.”) is not only the ML-based sensitivity enhancement but the **combination** of the enhancement with the nanofluidic reactors that indeed host a catalyst and are necessary if the sensitivity enhancement of the QMS is to be applied in catalysis. In other words, we argue that our work indeed makes a contribution to the field of catalysis as the combination of ML and nanofluidic reactor **together** enable the reduction of the required catalyst surface area for online QMS measurements. This reduction, in our opinion, indeed constitutes a relevant contribution to the field of catalysis science.

1. Chen, Y. (2015). Nanofabrication by electron beam lithography and its applications: A review. *Microelectronic Engineering*, 135, 57–72. doi:10.1016/j.mee.2015.02.042